# Restoration of patterned vision with an engineered photoactivatable G protein-coupled receptor

Michael H. Berry[1,2], Amy Holt[1], Joshua Levitz[1,10], Johannes Broichhagen [3,4,11], Benjamin M. Gaub[1,5], Meike Visel[1], Cherise Stanley[1], Krishan Aghi[6], Yang Joon Kim[7], Kevin Cao[1], Richard H. Kramer[1,6], Dirk Trauner[3,12], John Flannery [1,6,8] & Ehud Y. Isacoff[1,6,7,9]

Retinitis pigmentosa results in blindness due to degeneration of photoreceptors, but spares other retinal cells, leading to the hope that expression of light-activated signaling proteins in the surviving cells could restore vision. We used a retinal G protein-coupled receptor, mGluR2, which we chemically engineered to respond to light. In retinal ganglion cells (RGCs) of blind *rd1* mice, photoswitch-charged mGluR2 ("SNAG-mGluR2") evoked robust OFF responses to light, but not in wild-type retinas, revealing selectivity for RGCs that have lost photoreceptor input. SNAG-mGluR2 enabled animals to discriminate parallel from perpendicular lines and parallel lines at varying spacing. Simultaneous viral delivery of the inhibitory SNAG-mGluR2 and excitatory light-activated ionotropic glutamate receptor LiGluR yielded a distribution of expression ratios, restoration of ON, OFF and ON-OFF light responses and improved visual acuity. Thus, SNAG-mGluR2 restores patterned vision and combinatorial light response diversity provides a new logic for enhanced-acuity retinal prosthetics.

[1] Department of Molecular and Cell Biology, University of California, Berkeley, CA 94720, USA. [2] Department of Medicine, Oregon Health and Science University, Portland, OR 97239, USA. [3] Department of Chemistry, Ludwig-Maximilians-Universität München, and Munich Center for Integrated Protein Science, Butenandtstrasse 5-13, 81377 München, Germany. [4] Laboratory of Protein Engineering, Institut des sciences et ingénierie chimiques, Sciences de base, École Polytechnique Fédérale Lausanne, 1015 Lausanne, Switzerland. [5] Department of Biosystems Science Engineering, ETH Zürich, Mattenstrasse 26, 4058 Basel, Switzerland. [6] Helen Wills Neuroscience Institute, University of California, Berkeley, CA 94720, USA. [7] Biophysics Graduate Program, University of California, Berkeley, CA 94720, USA. [8] School of Optometry, University of California, Berkeley, CA 94720, USA. [9] Bioscience Division, Lawrence Berkeley National Laboratory, Berkeley, CA 94720, USA. [10]Present address: Department of Biochemistry, Weill Cornell Medical College, New York City, New York 10024, USA. [11]Present address: Department of Chemical Biology, Max-Planck Institute for Medical Research, Jahnstr. 29, 69120 Heidelberg, Germany. [12]Present address: Department of Chemistry, New York University, New York City, New York 10003, USA. Correspondence and requests for materials should be addressed to E.Y.I. (email: ehud@berkeley.edu)

nherited retinal degenerative diseases such as Retinitis pigmentosa (RP) cause progressive loss of rod and cone photoreceptors[1, 2,], leading to blindness[3], but spare downstream neurons of the inner retina for decades, providing a target for treatment[4, 5]. In hopes of restoring sight, a number of approaches have conferred light sensitivity to these remaining cells in animal models of RP, such as the *rd1* mouse, by expressing light-sensitive signaling proteins, including microbial ion channels and pumps[6–10], chemically engineered receptor-channels[11, 12] and photochemicals that sensitize native ion channels[13–17]. These light-sensors have been targeted to remnant cones, ON-bipolar cells (ON-BCs) or retinal ganglion cells (RGCs), endowing them with

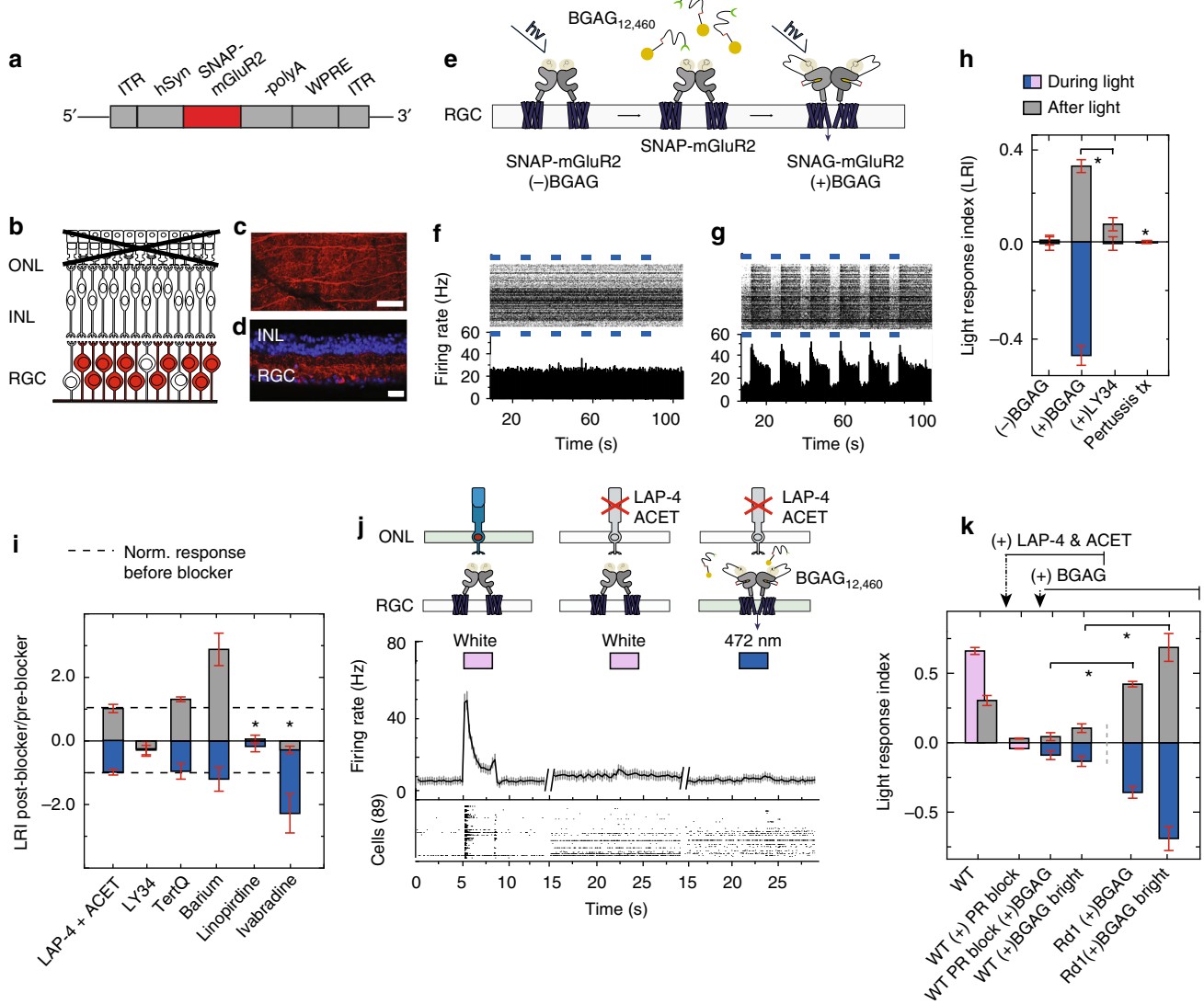

**Fig. 1** Expression of SNAG-mGluR2 in RGCs of *rd1* mouse retina. **a** Viral DNA expression cassette. SNAP-mGluR2 (red) under control of the *hsyn-1* promoter. **b** Schematic of a degenerated *rd1* mouse retina with targeted cells highlighted (red). ONL outer nuclear layer. IPL: inner plexiform layer. **c, d** Flat mount (**c**) and slice (**d**) confocal images of SNAP-mGluR2 expression in RGCs of *rd1* mouse retina 4 weeks after intravitreal injection of *AAV2/2-hSyn-SNAP-mGluR*. SNAP-Surface Alexa Fluor 647 dye (red) used to visualize SNAP-mGluR2 and DAPI (blue) to visualize nuclei. Scale of 60 and 20 μm. **e** Schematic of SNAP-mGluR2 labeling by BGAG$_{12,460}$ and photo-activation in RGCs. **f, g** MEA recordings from expressing *rd1* mouse retinas in the absence of photoswitch (**f**) or following labeling with BGAG$_{12,460}$ (**g**). (Top) Raster plot with spikes for each RGC (f:n = 120; g:n = 124). (Bottom) Peristimulus time histogram (PSTH). Light stimulation protocol: 5 × 5 s light (λ = 445 nm, blue bars) separated by 10 s dark. **h** Normalized Light response Index (LRI) for retina expressing SNAP-mGluR2 with no BGAG$_{12,460}$, after 45 min of BGAG$_{12,460}$, then in the presence of 5 μM LY341495 (N = 3, n = 156) and in separate retina LRI following retina injected with 150 μM pertussis toxin for 24 h (N = 4, n = 188). **i** Ratio of change in LRI before compared to after (LRI post-/LRI pre-) application of 50μM L-AP4 and 1 μM ACET (N = 2, n$^{channel}$ = 65), 5 μM LY341495LY (N = 3, n$^c$ = 73), 300 nM Tertiapin-Q (N = 2, n$^c$ = 42), 1 mM barium (N = 2, n$^c$ = 27), 500 nM linopirdine (N = 2, n$^c$ = 37), or 50 μM ivabradine (N = 2, n$^c$ = 46) in the recording solution. **j** Schematic (top) trace response to 3 s light pulse in *wt* retina expressing SNAG-mGluR2. White light response before (left) and after (middle) photoreceptor block via addition of 50μM L-AP4 and 1 μM ACET and minimal light response to 445 nm with BGAG$_{12,460}$ (right) (n = 98). **k** Normalized LRI of *wt* retina expressing SNAG-mGluR2 with photoreceptor blocker (before and after PR blocker = 50μM L-AP4 and 1 μM ACET) when illuminated by white light 100 μW cm$^{-2}$ (pink), 445 nm light (20 mW cm$^{-2}$) or 472 nm light (50 mW cm$^{-2}$). Response compared with *rd1* mice expressing SNAG-mGluR2. Light intensity 25 mW cm$^{-2}$, BGAG$_{12,460}$ labeling at 25 μM, N = # of retina, n = # of cells/units, n$^c$ = # of channels. All units refer to sorted cells. SEM in gray or as error bars. Statistical significance assessed using Mann-Whitney U test (*p ≤ 0.001) (Supplementary Table 1)

a uniform response to light. Though these approaches lack the diversity of the visual responses found in normal vision, most have succeeded in restoring light-evoked activity in visual cortex[18, 8], pupillary reflex[11, 14, 19], and discrimination of light from dark[7, 11, 12, 14, 16, 18–20]. It remains to be determined whether these systems support image recognition.

While most efforts have employed light-activated ion channels, light-activated G protein-coupled receptors (GPCRs), such as the opsins of photoreceptor cells, represent an attractive alternative being native to the retina and functioning with high sensitivity, and possibly at low expression, because they activate channels downstream of an amplifying signal cascade. Indeed, recently ectopic expression of rhodopsin or melanopsin was shown to restore light responses under dim light[18, 19, 21, 22]. Unfortunately, outside of photoreceptor cells, rhodopsin and melanopsin generate very slow light responses, likely preventing them from supporting vision during motion.

To obtain a native retinal GPCR that endows a rapid light response onto RGCs, we engineered metabotropic glutamate receptor 2 (mGluR2) to contain a nanoscopic chemical photoswitch that activates mGluR2 in response to light. We employed a new photoswitch attachment method, where the catalytic protein-tag SNAP is fused to the N-terminus of the target receptor (generating SNAP-mGluR2) and the photoswitch conjugates to the SNAP domain via a selective benzylguanine-reactive group[23]. We targeted SNAP-mGluR2 to RGCs, the cells that survive longest in RP[2–5] and so hold the broadest clinical potential. We found the photoswitch-charged "SNAG-mGluR2" produces a uniform OFF light response across the RGC population that is similar to that of OFF RGCs in the wild-type (wt) retina[24]. The SNAP-targeted photoswitch has both high attachment selectivity and very long functional longevity when injected in an FDA approved slow-release excipient—important advances for clinical application. Strikingly, SNAG-mGluR2 produced little or no light response in the RGCs of wt mice, suggesting that, in the degenerating retina, it may selectively act in RGCs whose upstream circuit has lost its photoreceptor cell input. In addition, using a new visually-guided image discrimination paradigm, we show that rd1 mice expressing SNAG-mGluR2 are not only capable of distinguishing light from dark, but can discriminate between static light patterns of equal luminance, a critical benchmark for vision restoration.

We asked whether diversification of the RGC light response, to more closely mimic the variable RGC responses in wt retina, would augment visual discrimination. To achieve this, we took advantage of the distinct methods of photoswitch attachment to generate the inhibitory (OFF responsive) SNAG-mGluR2 and the excitatory (ON responsive) LiGluR[11, 12]. We found that simultaneous transfection with adeno-associated viruses (AAVs) for these engineered receptors leads to RGCs that display a range of ON, OFF, and ON-OFF light responses. Strikingly, rd1 mice expressing this dual system performed better in close-line discrimination than their littermates that express either SNAG-mGluR2 or LiGluR alone. Our findings indicate that the OFF-responsive GPCR SNAG-mGluR2 restores vision for many weeks after a single photoswitch injection and supports patterned vision. SNAG-mGluR2 operates in RGCs that have lost their light response due to degeneration of their photoreceptor inputs, but not in wt retina, suggesting that in a retina with partial loss of photoreceptors—as in early stage disease—it will not interfere with RGCs that retain photoreceptor drive. Visual acuity is enhanced by diversification of the light response by combinatorial-randomized co-expression of SNAG-mGluR2 with a second ON-responsive receptor. The combinatorial approach provides an exciting design principle for enhanced vision restoration using an optogenetic retinal prosthetic.

## Results

**SNAG-mGluR2 restores a retinal light response.** For vision restoration, we turned to our chemically and genetically engineered light-activated version of mGluR2, a class C GPCR that is natively expressed in retinal neurons and implicated in the diverse modulatory effects of visual perception[25]. To photosensitize mGluR2 we used a Photoswitchable Orthogonal Remotely Tethered Ligand (PORTL) with a benzylguanine (BG) reactive group at one end, an azobenzene (A) photoisomerizable group in the center and a glutamate (G) ligand on the distal end (hence "BGAG")[23, 26]. The BG group irreversibly conjugates to a catalytic SNAP domain[27] that was fused to the N-terminus of mGluR2[28]. When SNAP-mGluR2 is "charged" by conjugation to BGAG it creates a photo-activated "SNAG-mGluR2"[23]. The BGAG-charged "SNAG-mGluR2" receptor was characterized in HEK293 cells (Supplementary Fig. 1) and found to have desirable characteristics for vision restoration (see Supplementary Note 1).

We tested SNAG-mGluR2 in the retina of the rd1 mouse after postnatal day 90, when rod and cone photoreceptor cells have degenerated[29]. SNAP-mGluR2 in an AAV vector under control of the synapsin promoter (hsyn-1) was packaged into virus, injected intravitreally and animals were tested ≥ 4 weeks later. To gauge SNAP-mGluR2 expression, we initially labeled the SNAP with a BG-conjugated Alexa Fluor-647 fluorophore (Fig. 1a–c). We found the expression to be pan-retinal (Supplementary Fig. 2a), specific for expressing retina (Supplementary Fig. 2b), restricted to the RGC layer, and covering the area where the somata and dendrites of both ON- and OFF-RGCs are located (Fig. 1d). This well-defined expression profile matched the expected targeting for this viral capsid[30], illustrating the great specificity of SNAP labeling with BG. In the remaining experiments labeling of SNAP-mGluR2 was always with the BGAG photoswitch.

SNAP-mGluR2 expressing retinas were removed from > 3 months old rd1 mice and mounted on a multi-electrode array (MEA) in order to test light evoked responses. Due to photoreceptor degeneration, no light-evoked responses were detected in absence of $BGAG_{12,460}$ (Fig. 1e, f). Following incubation of $BGAG_{12,460}$, large light-evoked responses were detected. These were observed in > 95% of cells identified by electrical activity (Fig. 1g, h), were consistent between retinas, and were similar to the natural OFF response reported in the OFF-RGCs of the wt retina[24, 31]. Rd1 retinal explants from animals that had not been injected with AAV-hSyn-SNAP-mGluR2, but which were incubated in $BGAG_{12,460}$ showed no light-dependent activity changes (Supplementary Fig. 2c, d), indicating that BGAG action depends on SNAP-mGluR2. Retinal response was normalized across cells and retina using a previously established Light Response Index (LRI)[12, 16, 17]. Application of 5 μM LY341495, a competitive antagonist of mGluR2[32], inhibited the light response (Fig. 1h and Supplementary Fig. 3a, b), as expected for photomodulation of firing that is driven by SNAG-mGluR2. Moreover, as expected for the Gi-coupled mGluR2[33], the light response was absent in retinas injected with 150 uM pertussis toxin for 24 h (Fig. 1h and Supplementary Fig. 3c). Furthermore, the addition of 50 μM L-(+)-2-Amino-4-phosphonobutyric acid (L-AP4; agonist of mGluR6[34]) and 1 μM (S)-1-(2-Amino-2-carboxyethyl)-3-(2-carboxy-5-phenylthiophene-3-yl-methyl)-5-methylpyrimidine-2,4-dione (ACET; blocker of kainate receptors[35]) to expressing rd1 retina had no effect, confirming that the observed light response is not reliant on any residual photoreceptors in the degenerative mouse (Fig. 1i and Supplementary Fig. 3d)

We next sought to identify the ion channel effector of SNAG-mGluR2 in the RGCs of the rd1 mouse. We considered that the inhibitory response could be mediated by activation of a potassium channel. We first tested G protein-coupled inwardly-rectifying potassium (GIRK) channels, since these channels are

expressed in RGCs[36] and are activated by $G_{\beta\gamma}$ as a result of activation of mGluR2[37]. However, neither 300 nM Tertiapin-Q nor 1 mM barium, both potent blockers of GIRK channels[38], abolished the SNAG-mGluR2 light response in the *rd1* retina (Fig. 1i and Supplementary Fig. 3e–g). We next considered Kv7 channels, since one of these, Kv7.4, is also activated by $G_{\beta\gamma}$[39, 40] and is expressed in the retina[41, 42]. We found that 500 nM linopirdine, a potent and, at this low concentration, selective blocker of Kv7 channels[43], blocked the SNAG-mGluR2 light response in the *rd1* retina (Fig. 1i and Supplementary Fig. 3h), suggesting that the SNAG-mGluR2 inhibitory light response is mediated by activation of the Kv7.4 channel. Lastly, we tested the hypothesis that the SNAG-mGluR2 OFF response is due to hyperpolarization-activated cyclic nucleotide-gated HCN channels, since HCN channels are expressed in RGCs[44] and have been established in their hyperpolarization-induced rebound

excitation[45]. Indeed, we found that 50 μM ivabradine, a blocker of HCN channels[46], eliminated the excitatory OFF response while maintaining the inhibitory component when the light was on (Fig. 1i and Supplementary Fig. 3i). Altogether, these results suggest that SNAG-mGluR2 activates Kv7.4 channels through Gi-linked Gβγ coupling to hyperpolarize RGCs and that deactivation of the channels when the light is turned off triggers rebound excitation due to activation of HCN channels.

RGCs in *wt* retina, in which photoreceptors are intact, spontaneously fire action potentials at a low basal rate in the dark[47]. In the *rd1* retina, the basal firing rate of RGCs increases by 3–10 times[48, 49]. Since light-activation of SNAG-mGluR2 inhibits RGC firing in *rd1* mice and induces a rebound OFF response, we wondered if SNAG-mGluR2 would have the same effect in the *wt* retina, which has lower levels of firing in the dark. SNAP-mGluR2 was virally delivered to RGCs of *wt* animals, leading to pan-

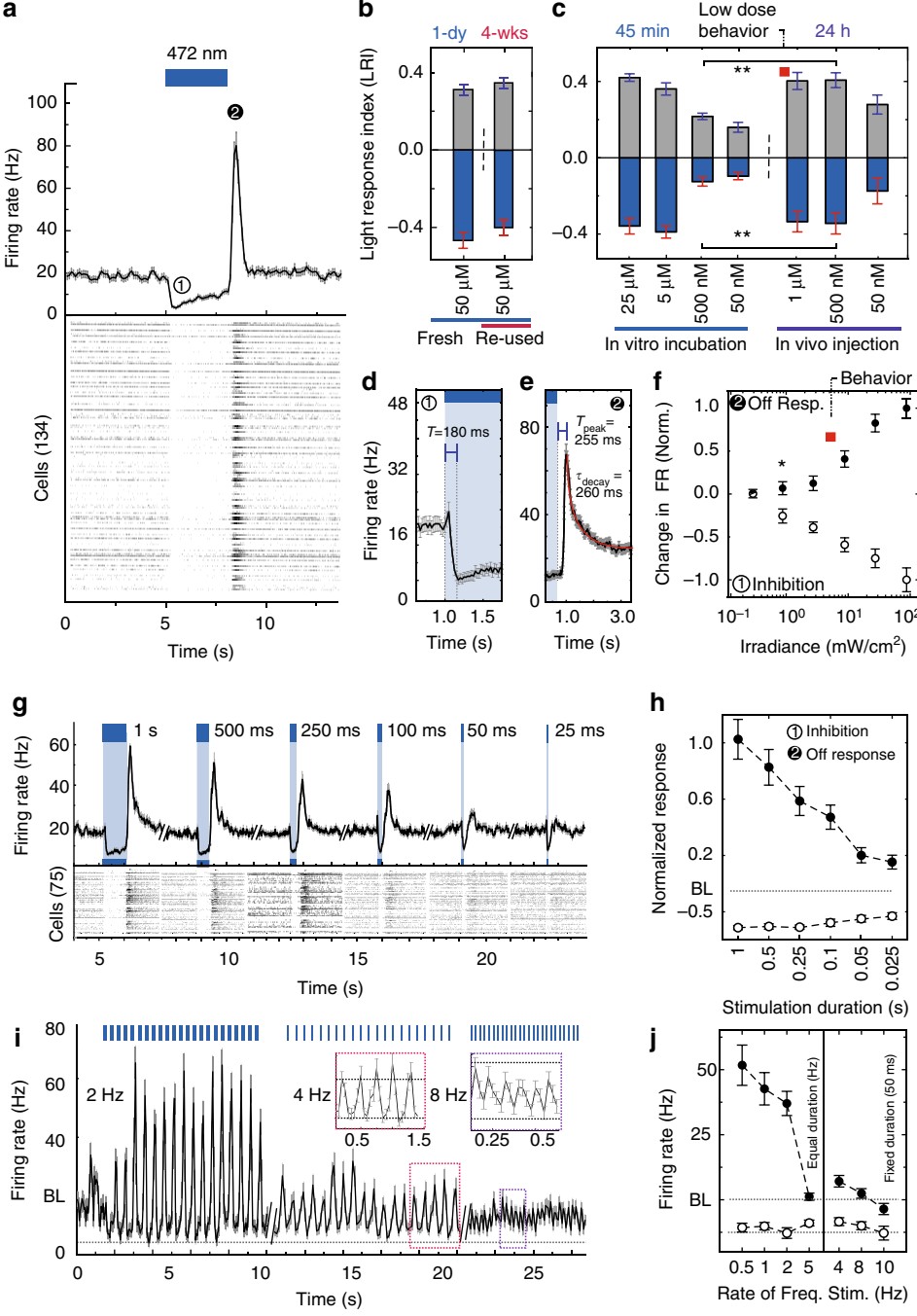

retinal expression, which was restricted to the RGC layer (Supplementary Fig. 4a–c), as seen in the *rd1* retina (Fig. 1d). Retinas mounted on the MEA and conjugated to BGAG$_{12,460}$ had low basal activity in the dark and normal photoreceptor mediated ON and OFF responses when flashed with dim white light that was too weak to activate SNAG-mGluR2 (Fig. 1j, k and Supplementary Fig. 3j). Block of synaptic transmission from photoreceptor cells to ON and OFF bipolar cells with 50 μM L-AP4 (agonist of mGluR6[34]) and 1 μM ACET (blocker of kainate receptors[35]), abolished this photoreceptor-driven light response (Fig. 1j, k and Supplementary Fig. 3k). Strikingly, illumination at the intensity (20 mW cm$^{-2}$) and wavelength (445 nm) that produces a large SNAG-mGluR2 light response in the RGCs of the *rd1* retina (Supplementary Fig. 3a) elicited little to no light responses in the *wt* retina (Fig. 2j). Increasing intensity by 2.5-fold evoked a small light response: ~7× smaller than that induced in *rd1* expressing animals (Fig. 1k and Supplementary Fig. 3l). These results suggests that the signaling of SNAG-mGluR2 occurs selectively in the degenerating retina, and that, in early stage disease, when only some of the photoreceptors have been lost, this therapy may selectively animate RGCs whose photoreceptor cells have degenerated, with minimal effect on RGCs whose photoreceptor cells remain intact.

Prior to our turn to SNAP-tag photoswitch attachment, photoswitches employed a maleimide reactive end to covalently attach to a cysteine introduced into the target receptor. Because maleimide hydrolyzes in water, these earlier photoswitches need to be injected into the vitreous at high concentration and lost reactivity in tens of minutes, limiting conjugation to a brief period after injection[11, 12]. In contrast, the BG reactive group of BGAG that attaches covalently to SNAP is stable in aqueous buffer[23]. Indeed, we found that BGAG$_{12,460}$ solutions stored at room temperature could be reused over 4 weeks with no decline in efficacy (Fig. 2a, b). Retinal explants expressing SNAP-mGluR2 in RGCs were sensitized to light by BGAG$_{12,460}$ concentrations as low as 50 nM, with maximal labeling consistently achieved at 5–25 μM (Fig. 2c), 4–20 fold lower than what is needed for maleimide photoswitches in vitro[12, 14, 16]. Intravitreal micro-injection at a final vitreal concentration of 500 nM also yielded a maximal light response the following day (Fig. 2c). We conjecture that this maximal efficacy is obtained at such a low concentration because of the protracted in vivo exposure time. This maximal efficacy in vivo concentration is ~2,000-fold lower than what is used for maleimide photoswitches[12, 14, 16] and ~40,000-fold lower than for non-covalent photoswitches such as DENAQ[16, 17]. Cell density in the *rd1* mouse retina was unaffected by repeated intravitreal injections of BGAG at a final vitreal concentration of 250 μm (Supplementary Fig. 2e). This indicates that the therapeutic window is at least ×500 (tolerated dose/dose required for maximal efficacy = 250 μM/0.5 μM).

MEA recordings showed that the light response of SNAG-mGluR2 in RGCs consists of two components. During illumination, firing was rapidly suppressed ("ON-inhibition"; $T_{inhib\text{-}onset}$ = 180 ± 7.0 ms) (Fig. 2a, d). Following illumination there was a transient burst of activity ("OFF-excitation") (Fig. 2a) that rose and decayed quickly ($\tau_{decay}$ = 260 ms ± 15.5 ms) (Fig. 2e). Time from light termination to peak response ($T_{peak}$ = 258.6 ± 5.2) was found to be close to that of photoreceptor derived OFF responses in the RGCs of *wt* retina ($T_{peak}$ = 176.55 ± 21.1)[50] The threshold photo-response was elicited at a moderate light intensity (0.5 mW cm$^{-2}$) (Fig. 2f), representing an ~100-fold improvement over halorhodopsin in RGCs, the prior optogenetic standard for restoration of the OFF response[10].

Response kinetics suggested that RGCs containing SNAG-mGluR2 should be able to follow a reasonably fast modulation of light intensity. Illumination with pulses as short in duration as 25 ms triggered a peak OFF excitation that ranged from 10 to 20% of the maximum, while the inhibition response was reduced to ~50% (Fig. 2g, h), an illumination-duration dependence similar to that of *wt* retina[51]. With trains of light flashes, responses declined as frequency increased, but detectable light modulation was still observed up to a frequency of 8 Hz (Fig. 2i, j). The kinetics of the SNAG-mGluR2 light response are ~5 and ~50-fold faster than rhodopsin[21] and melanopsin[19]. This makes SNAG-mGluR2 the fastest GPCR system for vision restoration in RGCs to date, rivaling the kinetic performance of light sensitive ion channels such as LiGluR[12], an incredible feat when considering the complex transduction that must occur.

## SNAG-mGluR2 restores innate light avoidance.

Having observed effective delivery of the BGAG$_{12,460}$ photoswitch at low concentration in vivo and fast light responses in the RGCs of isolated *rd1* retinas, we sought to determine if blind mice restored with OFF light responses could perform visual tasks. Mice innately avoid brightly lit spaces, a survival mechanism associated with evading capture[52]. This behavior is lost following photoreceptor degeneration in the *rd1* mouse[7, 19]. To determine if this behavior could be rescued, *rd1* mice expressing SNAP-mGluR2 were tested in a behavior box containing adjoining light and dark compartments, before and after intravitreal injection of BGAG$_{12,460}$ ($n = 7$). *Rd1* mice with only SNAP-mGluR2 or only BGAG$_{12,460}$ spent equal amounts of time in the light and dark compartments, a lack of preference consistent with blindness

**Fig. 2** Properties of light response in isolated *rd1* mouse retina with SNAG-mGluR2 in RGCs. **a** (top) Average response of RGC population showing peak inhibition (1) and OFF response following light termination (2). (bottom) Averaged raster plot ($n = 134$) $5 \times 3$ s duration with 472 nm light flashes. **b** BGAG$_{12,460}$ stable for weeks in solution. No significant difference in peak responses between SNAP-mGluR2 expressing *rd1* retinas labeled with freshly solubilized BGAG$_{12,460}$ ($n = 58$ cells) vs. BGAG$_{12,460}$ stored in aqueous solution for 4 weeks ($n = 46$) ($N = 2$). **c** BGAG$_{12,460}$ labeling responses appear maximal at 5 μM when bathing retina for 45 min and 500 nM when injected intravitreally in vivo 24 h before retinal isolation ($n > 50$ units per retina, $N = 14$). Low done behavioral experiments performed at concentration indicated by red square. **d, e** Time-course of light response. Population average traces with time from light onset to max inhibition (**d**), exponential fit for OFF response decay (red) and time from light termination to max excitation (**e**). $n = 109$ and 95, $N = 2$. Time to peak of *wt* OFF responses also measured for comparisons $n = 63$, $N = 2$. **f** Light sensitivity for SNAG-mGluR2 in RGCs of *rd1* mouse retina. Peak firing rates normalized for inhibition (open circle) and OFF response (closed circle). ($n = 100$, $N = 2$). Behavioral experiments performed at intensity indicated by red square. **g, h** Dependence of response on flash duration. **g** Representative retina light response ($n = 75$): individual cell response raster plot (top) and population average firing rate (bottom). ± SEM reveal detectable responses down to 25 ms duration flashes. **h** Average peak inhibition (open circle) and OFF response (filled circle) for different stimulation durations ($n = 139$, $N = 2$). **i, j** SNAG-mGluR2 enables RGCs to follow light pulses at up to 8 Hz. **j** Average peak inhibition ± SEM (open circle) and OFF response (filled circle) for different frequencies with either equal duration light pulses and dark intervals (0.5–5 Hz: 1 s light/1 s dark–100 ms light/100 ms dark) or fixed duration (50 ms) flashes with varying dark intervals (4–10 Hz) ($n = 120$, $N = 2$). Light intensity 25 or 50 mW cm$^{-2}$, Wavelength: $\lambda = 445$ nm, BGAG$_{12,460}$ labeling after retinal isolation for 45 min at 50 μM unless otherwise specified. $N = $ # of retina, $n = $ total units/cells. All units refer to sorted cells. SEM in gray or as error bars. Statistical significance assessed using Mann–Whitney $U$ test (*$p \leq 0.001$) (Supplementary Table 1)

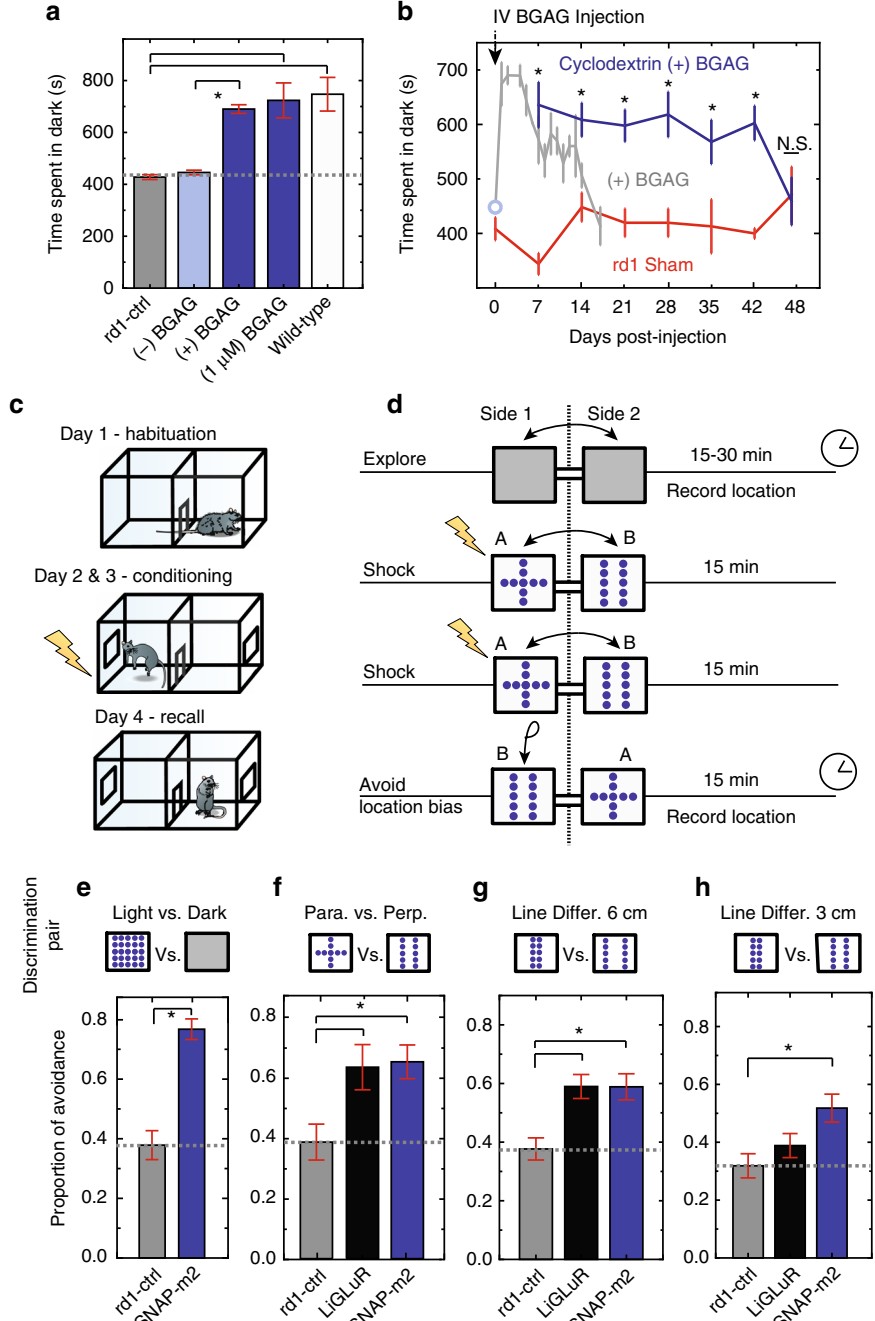

**Fig. 3** Light avoidance and learned visually guided behavior in *rd1* mouse expressing SNAG-mGluR2. **a** Restoration of light avoidance behavior. *rd1* mice (*n* = 7) and *rd1* mice expressing SNAP-mGluR2 spend equal times in dark and light compartments (grey dotted line) before delivery of BGAG₁₂,₄₆₀ (–BGAG) but prefer the dark compartment after intravitreal injection of BGAG₁₂,₄₆₀ ( + BGAG) (*n* = 7) to a similar level to wild-type mice (*n* = 7) and to mice treated with 500x lower photoswitch (*n* = 8). **b** Restored light avoidance persists 2 weeks after single injection of 1 mM BGAG₁₂,₄₆₀ (gray) (*n* = 7). When 3uM BGAG₁₂,₄₆₀ is combined with beta cyclodextrin in PBS as a method slow release drug delivery, light avoidance persists for 42 days (one-way ANOVA *p* < 0.005) with no decline in performance (blue) until day 48 (*n* = 11) (rANOVA *p* = 0.401). *Rd1* sham injected mice were assessed over the same 48-day period (red) (*n* = 8) (Supplementary Table 2). **c, d** Schematic of pattern discrimination experiment. Mice habituated at day 1, exposed to electric shock in association with specific pattern of light (stimulus A/B) paired randomly in either chamber on days 2 and 3 and tested (time spent in each chamber) on day 4, in absence of shock with light patterns reversed to avoid location bias. **e** Learned dark avoidance behavior. Proportion of time spent avoiding the dark after paired conditioning with shock. *rd1* SNAG-mGluR2 (*n* = 6), *rd1* control (*n* = 6). Values mean ± SEM **f–h** Learned pattern discrimination. Proportion of time spent avoiding pattern paired with shock. **f** Perpendicular vs. parallel bars. **g, h** Discrimination of parallel bars at distances of 1 vs. 6 cm (**g**) or 1 vs. 3 cm (**h**). Respectively for **f**, **g** and **h**: *rd1* control (*n* = 7, 13, 12 mice), *rd1* LiGluR (*n* = 7, 14, 15 mice), *rd1* SNAG-mGluR2 (*n* = 7, 10, 18 mice). In addition, proportion of success was also calculated (Supplementary Fig. 5b, d–g and Supplementary Table 3). All animals received 2 μL intravitreal injection of 1 μM BGAG₁₂,₄₆₀ in each eye and assayed following 24 h recovery. Display of 472 nm light equaled 5 mW cm⁻² at decision point. *n* = # of mice. Statistical significance was assessed using repeated-measures ANOVA (**b**), one-way ANOVA: *\*p* < 0.005 (**b**) (Supplementary Table 2), and Student's two-tailed *t*-test with Bonferroni correction: *\*p* < 0.01

(Fig. 3a). In contrast, the same group of expressing SNAP-mGluR2 *rd1* mice, when re-tested following intravitreal injection of BGAG$_{12,460}$, showed a marked preference for the dark compartment with avoidance similar to that of *wt* mice ($n = 7$), even when final vitreal photoswitch concentration was only 1 μM (Fig. 3a), confirming that the therapeutic window is large, i.e., at least ×250 (250 μM/1 μM).

A challenge for visual restoration with chemically-engineered receptors is their requirement for a synthetic photoswitch, which must be added periodically since the reactive form of the photoswitch persists for a limited time in the vitreous and the photoswitch-conjugated receptor turns over. In the case of LiGluR and its briefly reactive maleimide photoswitch, the light response lasts for about 1 week[12]. Following a single injection of 1 mM BGAG$_{12,460}$, we found light avoidance behavior to persist for ~2 weeks, with a half-life ($t_{1/2}$) of ~8 days (Fig. 3b). We wondered if the persistence could be extended if we took advantage of the great stability of the SNAP-reactive form of BG. To test this, we placed BGAG$_{12,460}$ in an FDA-approved excipient for intravitreal injection: pharmaceutical grade beta cyclodextrin in PBS. Intravitreal injection of BGAG$_{12,460}$ in cyclodextrin/PBS to yield a final vitreal concentration of 3 μM BGAG$_{12,460}$ led to light avoidance that persisted with no sign of decay for 42 days ($n = 11$) (Fig. 3b). This indicates that a clinically viable formulation can vastly extend the efficacy of vision restoration following a single photoswitch dose.

We next sought to determine if SNAG-mGluR2 in RGCs would enable *rd1* mice to perform active avoidance by conditioning to counter-intuitively associate the preferred dark compartment with an aversive stimulus. After a period of familiarization, mice are free to move between the dark and light compartments but are exposed to a foot shock whenever they venture into the dark side. During the recall phase, in which no shock is administered, the light and dark sides are flipped to account for location bias, and the time spent in each compartment is measured (Fig. 3c, d). We found that SNAG-mGluR2 *rd1* mice overcome their light-avoidance behavior, in contrast to control *rd1* mice (Fig. 3e), indicating an ability to associate the dark-light difference with the presence or absence of the noxious stimulus.

**SNAG-mGluR2 restores learned image discrimination**. We adapted the active avoidance behavior to an image discrimination task in which a custom LED panel mounted on each wall of the two compartments was programed to display a defined image of equal light intensity (3 mW cm$^{-2}$ measured from point of decision), wavelength (472 nm) and size (6 cm) but each displaying a distinct light pattern. For each animal an aversive foot shock was paired randomly with one of the two light patterns and used consistently during a training period (Fig. 3c, d). Upon recall, conditioned avoidance was tested. The patterns used were a pair of parallel (∥) lines or a pair of perpendicular (+) lines. LiGluR, a previously established light-activated ionotropic glutamate receptor[12], was also tested as a point of comparison. We found that animals with either SNAG-mGluR2 ($n = 13$) or LiGluR ($n = 5$) in RGCs showed strong avoidance of the aversive stimulus (Fig. 3f), and that discrimination was similar to that seen in *wt* mice ($n = 5$) (Supplementary Fig. 5a). *Rd1* control animals predictably displayed a preference for the location not associated with the shock, with no regard for the stimulus, a clear example of location bias ($n = 6$). SNAP-mGluR2 mice also showed successful discrimination behavior when exposed to a very low vitreal concentration of 1 μM BGAG$_{12,460}$ ($n = 7$) (Supplementary Fig. 5b), confirming the specificity and stability of the photoswitch. In support of this analysis, the proportion of successful avoidance of pattern paired with shock was also determined for all learned discriminations behaviors (Supplementary Fig. 5c–g and Supplementary Table 3) (see Methods).

We next turned to an assessment of visual acuity using a pair of parallel lines separated by distances of 1 vs. 6 cm, or 1 vs. 3 cm

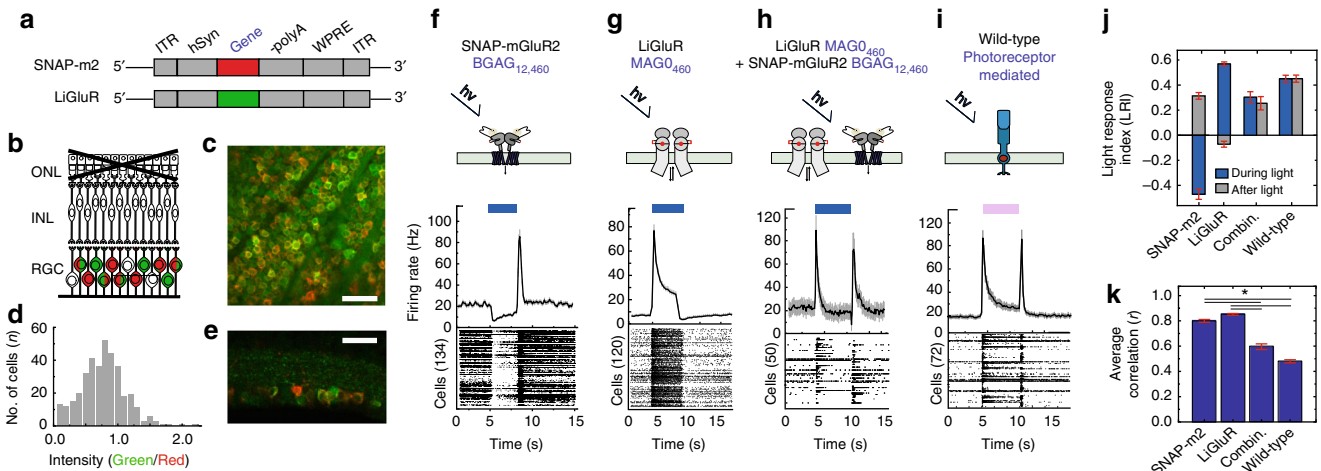

**Fig. 4** Light responses when combining LiGluR and SNAG-mGluR2 in RGCs. **a** Viral DNA expression cassette. SNAP-mGluR2 (red) and LiGluR (green). **b–e** RGCs expressing LiGluR (green) and SNAP-mGluR2 (red). **b** Schematic of degenerated *rd1* mouse retina. **c, e** Confocal images of *rd1* mouse retina 4 weeks after intravitreal injection of 1:1 mixture of *AAV2/2-hSyn-LiGluR* and *AV2/2-hSyn-SNAP-mGluR2* ($5 × 10^{11}$ viral genomes). LiGluR stained with anti-iGluR6 (green); SNAP-mGluR2 with BG-Alexa Fluor 647 (red). **d** Histogram displaying the distribution of anti-iGluR6 (green)/SNAP-mGluR2 (red) intensity ratios for each cell $n = 354$, $N = 3$. **f–i** Schematic (top) and RGC MEA response to 5 s pulse of 472 nm light (bottom) in *rd1* retina whose RGCs contain either SNAP-mGluR2 + BGAG$_{12,460}$ (**f**), LiGluR + MAG0$_{460}$ (**g**), or both SNAP-mGluR2 and LiGluR + photoswitchs (**h**). Photoreceptor-mediated response in *wt* shown for comparison (**i**). Raster plots of individual units are averages of 5 flashes, SEM in gray. **j** LRI of peak responses during (blue) and after (gray) illumination in *rd1* retinae with SNAG-mGluR2, LiGluR, SNAP-mGluR2 and LiGluR together (Combin.), and in *wt* retina. Values mean ± SEM $n \geq 50$, $N = 4$. **k** Average cross-correlation values in *rd1* retinas with SNAG-mGluR2 ($n = 134$), LiGluR ($n = 120$), SNAG-mGluR2 + LiGluR ($n = 90$), and in wild-type retina ($n = 93$). Cross-correlation of all light-sensitive units in period 1 s before to 2 s after light pulse. Values mean ± SEM. Light 472 nm at 50 mW cm$^{-2}$. Photoswitch concentration 50 μM. $N = $ # of retina, $n = $ # of units/cells, all units refer to sorted cells. Statistical significance was assessed using Mann-Whitney $U$ test (*$p < 0.001$) (Supplementary Table 1)

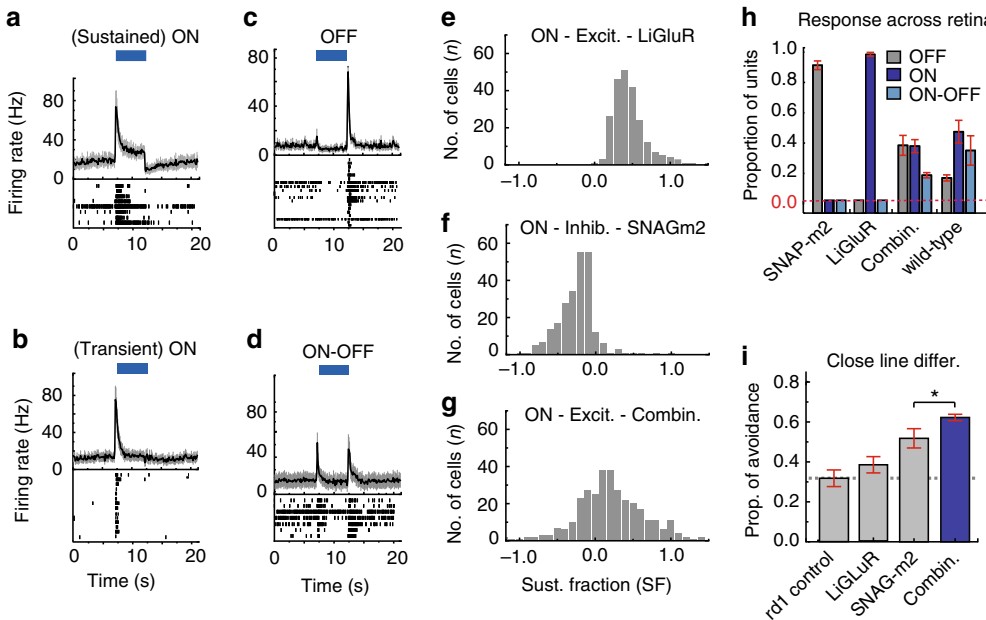

**Fig. 5** Assessment and behavioral function with restoration using co-expression of LiGluR with SNAG-mGluR2. **a–d** (top) Averaged response of RGC ON-sustained (**a**), ON transient (**b**), OFF (**c**), ON–OFF (**d**) responding population within the same retina expressing both SNAG-mGluR2 and LiGluR. Standard errors in gray. (Bottom) Averaged raster plot of individual units for 10 flashes of blue light. **e–g** Histograms showing distribution across RGC population in sustained fraction of the light response, calculated as the firing rate at the end of illumination divided by the peak-firing rate for each cell (larger peak response selected between initial response at the start of illumination and the rebound OFF response). $N = 3$ retinas per condition: LiGluR alone ($n = 227$); **f**), SNAG-mGluR2 alone ($n = 246$) and SNAP-mGluR2 + LiGluR co-expression ($n = 222$). **h** Proportion of units identified in retinas that display ON (blue), OFF (gray) and ON-OFF (light blue) response to full field illumination. Retinas expressing SNAP-mGluR2 (Left), LiGluR (Middle-left), both SNAP-mGluR2 and LiGluR (Middle-right), or wt ($N = 3$ retinas per condition). Error bars represent S.D., red dashed line denotes zero, emphasizing the absence of ON and ON-OFF responses in SNAG-mGluR2 alone and of OFF and ON-OFF responses in LiGluR alone. **i** Learned pattern discrimination behavior using paring of parallel bars with distances of 1 vs. 3 cm. Plotted mean proportion of time spent on non-aversive side (avoiding) the pattern associated with the shock for rd1 co-expressing LiGluR + SNAG-mGluR2 (Blue) ($N = 18$). rd1 untreated control (Gray-Left) ($n = 12$), LiGluR (Gray-Middle) ($N = 15$), rd1- SNAP-mGluR2 (Gray-right) ($N = 18$), reproduced from Fig. 4h for reference and comparison. In addition, proportion of success was also calculated (Supplementary Fig. 5 b, d–g and Table 3). Light intensity 25–50 mW cm$^{-2}$, BGAG$_{12,460}$ labeling at 50 μM. $N =$ retina or mice per condition, $n =$ total units/cells. All units refer to sorted cells. All animals received 2 μL intravitreal injection of 1 μM BGAG$_{12,460}$ in each eye and assayed following 24 h recovery. Display of 472 nm light equaled 5 mW cm$^2$ at decision point. Statistical significance was assessed using Student's two-tailed t-test: *$p < 0.05$ (**i**)

line separation, with an aversive foot shock paired with a randomly selected stimulus. In the first case, with the larger difference (6 cm), rd1 mice with SNAG-mGluR2 ($n = 10$) distinguished between stimuli equally well to those expressing LiGluR ($n = 10$) (Fig. 3g and Supplementary Fig. 5f, Supplementary Table 3). When the difference in distance was reduced (3 cm), LiGluR ($n = 13$) animals did not perform significantly better than untreated animals ($n = 12$), but SNAG-mGluR2 ($n = 18$) animals successfully discriminated between the stimuli (Fig. 3h and Supplementary Fig. 5g, Supplementary Table 3), as did wt animals ($n = 11$) (Supplementary Fig. 5a, c, Supplementary Table 3). These results indicate that in RGCs SNAG-mGluR2 provides higher acuity discrimination than LiGluR.

**Co-expression restores diverse retinal responses.** Having seen that the inhibitory (OFF-responsive) SNAG-mGluR2 and excitatory (ON-responsive) LiGluR can each support visual discrimination, we wondered if combining these systems could reconstitute a greater response complexity in RGCs and further enhance visual acuity. We co-injected a 1:1 ratio of identical AAVs packaged with either the SNAG-mGluR2 or LiGluR gene (Fig. 4a, b). Expression was visualized > 4 weeks later, following sequential labeling of SNAP-mGluR2 with the BG-conjugated Alexa Fluor-647 and LiGluR with an Alexa − 488 anti-GluK2 antibody (Fig. 4c–e). SNAG-mGluR2 and LiGluR each showed

pan-retinal expression (Fig. 4e). The staining varied from cell to cell (Fig. 4d, e), with cells ranging from predominantly SNAG-mGluR2 (red), to predominantly LiGluR (green) to varying mixtures (shades of yellow) (Supplementary Fig 6a–d), suggesting that some RGCs would behave like OFF or ON cells, while others may have mixed properties.

Retinas co-expressing SNAP-mGluR2 and LiGluR in RGCs were excised and placed on the MEA. BGAG$_{12,460}$ and MAG0$_{460}$ were applied and light-evoked activity of was compared with that of retinas expressing either SNAG-mGluR2 or LiGluR alone. As established, SNAG-mGluR2 alone yielded a uniform OFF response in the RGCs, whereas LiGluR alone yielded a uniform ON response (Fig. 4 f,g). In contrast, co-expression of SNAG-mGluR2 and LiGluR yielded an ON and OFF type response that resembled that observed in wt retinas (Fig. 4h–j).

Co-expressing retinas displayed an average cross-correlation index ($r = 0.59 + 0.02$) approaching that of wt ($r = 0.48 \pm 0.01$), and substantially lower than observed in retinas expressing either LiGluR alone ($r = 0.85 \pm 0.07$) or SNAG-mGluR2 alone ($r = 0.80 \pm 0.01$) (Fig. 4k and Supplementary Fig. 7, Supplementary Table 1). An examination of light responses from individual units revealed that lower correlation in the co-expressing retinas arises from a diversity of responses between cells (Fig. 4h). The responses could be subdivided into four groups (see Methods): ON sustained, ON transient, ON-OFF and OFF, (Fig. 5a–d). While some of the RGCs in the co-expressing retinas had light

responses similar to that observed in either LiGluR or SNAG-mGluR2 when expressed alone (compare Fig. 4f and g to Fig. 5a and c), the majority of units displayed unique intermediate light responses (Fig. 5a–d). To quantify this diversity we measured the sustained fraction (SF) of the response (see Methods). The RGCs of retinas with LiGluR alone had a range of positive SF values (Fig. 5e), whereas those with SNAG-mGluR2 alone had a range of negative SF values (Fig. 5f). Retinas that co-expressed SNAG-mGluR2 and LiGluR displayed a broad range of SF values that covered these individual ranges and peaked at an intermediate value (Fig. 5g). The wider range of ON, OFF and ON-OFF light responses made retinas co-expressing SNAG-mGluR2 and LiGluR more closely resemble the distributions seen in the of wt retina (Fig. 5h).

We confirmed that this novel diversity was due to the combined activity of SNAG-mGluR2 and LiGluR by applying 5 µM of the mGluR antagonist LY341495 to co-expressing retinas. We found that ON-transient units developed a sustained component and ON-sustained units developed a larger sustained component, with no effect on the LRI (Supplementary Fig. 8a–e). Thus, the varied intermediate light responses are attributed to the combined operation of SNAG-mGluR2 and LiGluR.

**Co-expression improves visual function**. Having observed greater diversity in RGC light responses, we asked if this combination would result in an advantage for vision. Using our image discrimination paradigm, we found that SNAG-mGluR2 and LiGluR co-expressing rd1 mice ($n = 18$) performed better in close line differentiation than did mice expressing either SNAG-mGluR2 ($n = 18$) alone or LiGluR alone ($n = 15$) (Fig. 5i and Supplementary Fig. 5g, Supplementary Table 3). This suggests that endowing a blind animal with a diverse RGC light response including both ON and OFF results in restored vision with enhanced acuity.

## Discussion

We demonstrate the translational potential for retinal gene therapy of a light-gated GPCR engineered from a metabotropic glutamate receptor. When virally targeted to the RGCs, "SNAG-mGluR2" introduces a fast OFF light response to these retinal output cells of the blind rd1 mouse model of RP. This OFF response supports pattern discrimination and restores higher visual acuity than does the ON response-generating LiGluR.

The photoswitch attachment chemistry that we employed is extremely selective for SNAP and is not subject to hydrolysis[53], enabling precise restriction to RGCs and maximal efficacy at a dose that is 500-fold lower than a dose that showed no sign of toxicity, indicating a very large therapeutic window. In contrast, the hydrolysable maleimide-targeted photoswitched tethered ligands, as used for LiGluR, and the non-covalent photopharmacological agents that target native ion channels, require 3–4 orders of magnitude higher concentrations[11–14, 16, 23]. The unique photoswitch stability in aqueous solution makes this system compatible with the FDA-approved excipient beta cyclodextrin in PBS, yielding persistence for ~6 weeks after a single low-dose injection, resolving the only significant barrier to optogenetic systems that employ a synthetic photoswitch.

Experiments show that SNAG-mGluR2 does not induce light responses in the wt retina at the physiological light intensities used in rd1. The mechanism for this selectivity in response may depend on the increase in baseline firing seen in rd1 animals[47, 48]. In addition, the reported increase in both specific Kv7 channel activity and up-regulation of HCN channels within the degenerating retina may contribute to selective function in the diseased retina[17, 54, 55]. Regardless, our results suggests that, in patients with early stage disease, it may be possible for SNAG-mGluR2 to selectively re-animate RGCs that have lost their light input without affecting RGCs that are still driven by functional photoreceptor cells.

SNAG-mGluR2 enabled blind rd1 mice to discriminate between parallel and perpendicular bars, and between parallel bars with 1 vs. 6 cm spacing, performance comparable to wild-type and rd1 mice expressing LiGluR. Strikingly, when the close-line discrimination task was made more difficult (distances of 1 and 3 cm), LiGluR animals were unable to perform the task, but the SNAG-mGluR2 animals successfully discriminated, indicating that this OFF responsive GPCR provides superior visual acuity.

SNAG-mGluR2 elicited an OFF response across the RGC population of the retina, whereas LiGluR in RGCs elicited an ON light response across the RGC population. Thus, even though there are as many as 30 different RGC subtypes[56], each light-gated receptor imposed a single class of light response and this was sufficient to restore an important aspect of patterned vision. However, in the normal retina, ON and OFF pathways combine to provide a signaling complexity that is thought to contribute to contrast detection and visual acuity[57]. We asked, therefore, whether by combining these systems we could reconstitute some of the response complexity of the wild-type retina[58] and thereby enhance pattern recognition. We did this by harnessing the stochastic nature of AAV-mediated expression and combining AAV-SNAP-mGluR2 with AAV-LiGluR to obtain a combinatorially randomized expression ratio that yields a wide variety of light responses across the RGC population. Functionally, this resulted in a broad distribution of light responses, ranging from cells with OFF responses resembling SNAG-mGluR2, ON-sustained responses resembling LiGluR alone, as well as transient ON and ON-OFF cells that ranged widely in behavior.

With this artificial restoration of the ON and OFF pathway we asked whether the brain would be able to take advantage of the more diverse information from the co-treated retina in order to improve visual acuity. Strikingly, rd1 mice that co-expressed SNAG-mGluR2 and LiGluR performed better than did those with either SNAG-mGluR2 alone or LiGluR alone in close line discrimination between distances of 1 and 3 cm. Accounting for distance from the point of decision and the dimensions of our patterned stimuli, this requires discrimination of ~9° of visual angle. Based on the Snellen value conversion for visual acuity that is commonly used to evaluate human eye sight[59], blind rd1 mice expressing SNAG-mGluR2 in RGCs have a resolving capacity of ~20/400 vision and those expressing the combination of SNAG-mGluR2 and LiGluR have a resolving capacity of ~20/200 vision. Given that the mouse retina lacks a fovea and resembles the peripheral retina of primates, this represents an impressive level of acuity.

The improved acuity that emerges from the combination of ON and OFF sensors suggests a new logic for retinal prosthetics that should be applicable to other pairs of optogenetic actuators. In this regard, co-expression of the microbial opsins Channelrhodopsin and Halorhodopsin has been used to recapitulate wavelength dependent ON and OFF light responses within the same retinal cell[10, 20, 6, 60]. However, differences in spectral and irradiance sensitivity[6, 10] make this pairing less ideal for restorative therapy, as does their requirement for very intense light. The success of the combinatorial approach suggests that there may be a general benefit to restoring a diversity of response properties to the blind retina, providing a novel design principle for enhanced vision restoration.

## Methods

**Animals, AAVs and photoswitches**. Mouse experiments were conducted under the express approval of the University of California Animal Care and Use

Committee. Equal numbers of male and female *wt* mice (C57BL/6 J) and equal numbers of male and female *rd1* mice (C3H) were purchased from the Jackson Laboratory (stock# 000659) and housed on a 12-h light/dark cycle with food and water ad libitum. cDNA encoding SNAP-mGluR2 was inserted in an established viral cassette under control of the human synapsin promoter (*hsyn-1*) and packaged in the AAV 2/2-4YF capsid. The vector, containing $10^{10}$–$10^{12}$ viral genomes was delivered in a 2 μl volume to the vitreous of the *rd1* mouse eye via micro-injection. rAAV injections were at p30–p60 and in vivo and in vitro experiments at p90–p160. AAVs were produced as previously described[61].

**Electrophysiology and light stimulation**. HEK cell recordings we performed using the methods previously established[23, 26]. MEA recordings were performed on *wt* (C57BL/6 J) mice, and untreated and treated *rd1* mice at > p90 6–10 weeks following AAV injection experimental retina were excised from the eye under dim red light, mounted on 4 μm cell membranes and placed in an incubator (35 °C) for 30 min. Retinal tissue was placed ganglion cell side down[62] in the recording chamber (pMEA 100/30iR-Tpr; Multi Channel Systems) of a 60-channel MEA system with a constant perfusion of Ames recording media (32 °C). A Multi Channel Systems harp weight was placed on the retina to prevent movement and vacuum was applied to the retina using a pump (perforated MEA1060 system with CVP; Multi Channel Systems), improving electrode-to-tissue contact and to provide consistent signal-to-noise ratios across retinas (10–20 Hz spontaneous activity). Further detail regarding MEA methods are previously detailed in Gaub et al.[12]. BGAG$_{12,460}$ or MAG0$_{460}$ were added to the recording chamber for 45 or 20 min, respectively, then perfused thoroughly to wash unbound photoswitch. For co-expressing retinas BGAG$_{12,460}$ was applied first. Illumination in vitro was by two light sources coupled to a 4 × objective: 1) a 300-W mercury arc lamp (DG-4; Sutter Instruments) 445/50 nm bandpass filter for BGAG$_{12,460}$ or a UV (380/15 nm) and visible (510/25 nm) bandpass filter for BGAG$_{12}$ or white light (no bandpass filter) and (2) an LED with collimator lens (472 nm, 25.0 mW cm$^{-2}$ or $6.3 \times 10^{16}$ photons per cm$^{-2}$·s$^{-1}$; Thorlabs, Inc.). Light intensity was controlled by modifying the light source duty cycle or by using neutral density filters and ranged from 0.45 to 50 mW cm$^{-2}$.

**Photoswitch preparation**. Photoswitch compounds were synthesized using the protocol described in Broichhagen et al.[23]. Stock solution of 200 mM BGAG$_{12,460}$ (L-diastereomer) in 100% pharmaceutical grade DMSO (Cryoserv; Bioniche Pharma) was diluted 1:100 in sterile PBS for a final working solution of 1 mM in 1% DMSO. Working solutions were either prepared before administration, prepared in stock stored in the freezer and used as required, or salvaged from the recording bath and stored (either RT or freezer) for reuse. Application of BGAG$_{12,460}$ or BGAG$_{12}$ on retinal explants were performed in a volume of 200 μL at a concentration of 50 μM to 50 nM BGAG$_{12,460}$ (in PBS with > 1% DMSO). For in vivo behavioral experiments, a 2-μL volume of 1 mM or 3.5 μL (final vitreal concentration of 1 μL) of BGAG$_{12,460}$ solution (in PBS with 1% DMSO) was injected into eyes that treated with AAV > 6 weeks earlier). For in vivo concentration dependence experiments, the mouse eye was assumed to contain a volume of 5.3 μL and a 2-μL volume of 3.5 μM, 1.825 μM and 0.1825 μM BGAG$_{12,460}$ solution (in PBS with 1% DMSO) was injected into eyes to obtain a final concentration of 1 μM, 500 nM and 50 nM[63]. For hydrated slow release 5% pharmaceutical grade beta cyclodextrin (*cyclodex*) in PBS was mixed with BGAG$_{12,460}$ for a final concentration of 3 μM and 2 μL were injected bilaterally into the mouse eye. MAG0$_{460}$ was synthesized and administered in 2 μL at a concentration of 100 μM, identical to the protocol established previously by Kienzler et al.[64] and Gaub et al.[12].

**Pharmacology**. In effort to identify the mechanism of SNAG-mGluR2 signaling in RGCs, we applied either 5 μM LY341495 (Tocris) or 300 nM Tertiapin-Q (Abcam) or 1 mM barium (Sigma) or 500 nM linopirdine (Sigma) or 50 μM ivabradine (Sigma) into the MEA recording bath. Additionally, concentrations of 150 uM pertussis toxin (PTX) were injected into the eye, 24 h later the eyes were removed and recorded from using the procedures described above. For *wt* retina expressing SNAP-mGluR2, the addition of 50 μM L-AP4 (Sigma), and and 1 μM ACET (Tocris) was applied to the perfusion in order to block photoreceptor mediated responses.

**MEA data acquisition and analysis**. Retinal activity on the MEA was sampled at 25 kHz filtered between 100 and 2,000 Hz and recorded using MC_rack software (Multi Channel Systems). Voltage traces were converted to spike trains offline and the spikes recorded at each electrode were sorted into single units, which we defined as "cells," via principal component analysis using Offline Sorter (Plexon-64bit) with each electrode commonly identifying 1–3 cells. Single-unit spike clusters were exported to MATLAB (MathWorks) and were analyzed and graphed with custom software. All firing rates were extracted from traces averaged over 5–30 light response cycles, details of which are denoted in figure legends. Responses across cells and across retina were normalized using the Light Response Index (LRI) adopted from Tochitsky et al.[16] and Gaub et al.[12] (LRI = (peak ON or OFF firing rate - average firing rate in dark) / peak ON or OFF firing rate + average firing rate in dark). Under experiments where conditions were changed within

retina (light sensitivity, dependence of response on flash duration, and frequency stimulation) the responses were normalized to the peak of the greatest response from baseline. Cells were defined as "responders" if the LRI satisfied the condition LRI > 0.1 or LRI < −0.1. Since there will be a floor effect when spike responses hit zero, while the cell membrane potential may be further hyper-polarized, our measurement is expected to underestimate the degree of inhibition in some conditions. Drug experiments comparing the change in LRI before and after application were calculated using a ratio LRI post-blocker/LRI before blocker for each channel, with increases in responses being > 1 (for OFF) or −1 (for inhibition) and decreases in responses being < 1 (for OFF) or −1 (for inhibition). Transients of cell response character was determined using Sustained Fraction (SF = steady state firing rate of each cell during illumination / peak response (ON or OFF)) at the beginning of illumination. Distinction of cell type response defined as follows: ON-sustained = LRI at light on > 0.1, LRI during light on > 0.1. ON-transient = LRI at light on > 0.1, LRI during light < 0.1. OFF = LRI at light on < 0.1, LRI at light OFF > 0.1. ON-OFF = LRI at light on > 0.1, LRI at light OFF > 0.1 at light on > 0.1. Correlations between cells were constructed by comparing the response character of peristimulus time histograms (PSTH) (1-s before illumination to 2 s following light termination) for each cell with one another using custom MATLAB software. Correlation values ranged between 1 and ~0 and a heat map was used to represent the correlation value of each data point in the matrix, with warmer colors indicating higher correlation values.

**Statistics and data**. To assess statistical significance, nonparametric two-tailed Mann-Whitney *U* tests where applied to the data when applicable (Supplementary Table 1). Statistical significance calculations for slow release BGAG delivery during behavioral light avoidance was analyzed by repeated-measures ANOVA (rANOVA) using "time point" as the within-subject variable and "group" (control untreated, treated) as the between-subject variable. Within-subject effects were analyzed by one-way ANOVA using "time point" as the independent variable. Where sphericity was violated, as assessed by Maulchy's test of sphericity, the Greenhouse–Geisser correction was applied (Supplementary Table 2). For learned dark avoidance behavior and the learned pattern discrimination behaviors, a standard deviation was computed. A success was defined as greater than the sum of the control group average and one S.D, and a failure was any value that did not achieve this criteria. Success ratios were then calculated for each condition (Supplementary Fig. 5c, d–g). To determine significance in differences between conditions a pairwise contingency table was then constructed, and a Two-Sided Pearson's Chi-Square Test was initially conducted. To correct for conditions with a small *n*, a One-Sided Fisher's Exact Test was also conducted (Supplementary Table 3). In addition, significance for behavioral performance (Fig. 3e–h and Fig. 5i) was also calculated using two-tailed unpaired Student's *t*-tests with Bonferroni correction.

**Tissue preparation and immunohistochemistry**. Mice > 4 weeks post-*AAV2/2-hsyn-SNAP-mGluR2* treatment were injected with 1 μL of 10 uM of BG-conjugated Alexa Fluor-647 dye into the vitreous. 25 h later mice were sacrificed, eyes were fixed in 4% paraformaldehyde (Ted Pella) (30 min), retinas were removed and the tissue incubated in blocking buffer (10% normal goat serum, 1% BSA, 0.5% Triton X-100 in PBS (pH 7.4)) for 2 h at RT. Retinas were washed thoroughly using PBS and flat mounted on slides using Vectashield (Vector Laboratories) medium impregnated with DAPI (cell nuclei stain - blue). Retinas additionally co-injected with *AAV2/2-hsyn-LiGluR* were exposed to monoclonal antibody against Anti-GluR6/7 (Millipore - Cat# 04–921) (1:500 dilution in blocking buffer overnight at 4 °C) and followed by secondary anti-rabbit Alexa 488 antibody (Invitrogen - Cat# A-11034) was applied (1:1000 dilution for 2 h at RT) previously described in Gaub et al.[12]. In vitro sequential labeling of SNAP-mGluR2 with the BG-conjugated Alexa Fluor-647 and antibody staining of the GluK2 subunit recognized in LiGluR was also successfully achieved using minimal fixation (10 min). For retinal sections, whole mounts were embedded in agarose (Sigma) and sectioned transverse using a vibratome (Leica Microsystems) at medium speed, maximum vibration, and 200-μm thickness. Retinal tissues used for immunohistochemistry on retinal cryosections or whole mounts were processed and examined by confocal microscopy (Leica TCS SP5; Leica Microsystems). For cell counting, retina was cryo-sectioned and stained with DAPI. Z-stack images (24 slices) of 1 μm$^3$ were obtained using the Zeiss LSM-880 NLO Airyscan microscope with ×20 objective, increased offset was used to minimize background and differentiate distinct cells, and analysis was performed using the Imaris software to count individual cells in the 3D image.

**Passive avoidance open field test**. The open field test was performed, as described previously[12, 19]. Briefly, a two-compartment (light and dark) shuttle box (Colbourn Instruments) allows the mouse to move freely through a small opening that connects the two compartments. The light compartment was illuminated by a custom LED array (6 × 6 LEDs, 447.5-nm Rebel LED; Luxeon Star LEDs) centered over the compartment. The light intensity (5 mW cm$^{-2}$ at floor level) was homogeneously distributed throughout the floor. Day 1 - Mice were transferred into the testing box, and allowed to habituate to the new environment with their littermates for 45 min. Mice were then retuned in to their home cage and then tested individually. Day 2 - Mice were placed in the light compartment and were given a

maximum of 3 min to discover that there is a second compartment. A 15-min trial began when they crossed into the dark compartment, and time spent in the light was recorded. Mice that crossed the opening only once and stayed in the dark compartment for entire time were disqualified. Animals are tracked using IR sensors on the shuttle box. Time spent on either side was collected and analyzed using the Graphic State, and Graphic State RT programs from Colbourn Instruments. $Rd1$ mice injected > 6 weeks earlier with $AAV2/2$-$hsyn$-$SNAP$-$mGluR2$ were test before and after intravitreal injection of $BGAG_{12,460}$. The same cohort was then additionally retested for light avoidance over a period of 2 weeks. $Wt$ mice were tested under identical conditions.

**Modified active avoidance and forced choice protocol.** The same two-compartment shuttle box (Colbourn Instruments) and illumination arrangement from active avoidance task was implemented in the modified active avoidance task. Mice are free to move between the light and dark compartment but received a 2 s, 0.7 mA shock when entering the dark compartment. In the forced choice task, a $6 \times 6$ LED array was mounted to the wall of each compartment. Each presented a different static LED pattern of equal light intensity (3 mW cm$^{-2}$ measured from point of decision), wavelength (470 nm - Digikey) and size (6 cm). Mice were free to move between the two compartments but one LED pattern was randomly parried with a shock (aversive) creating a preference for LED pattern not paired with the shock (non-aversive) in animals with visual perception. All equipment was treated with alcohol (training) or mild soap solution (testing day) between animals and cages. Prior to animal entering any equipment, cage is treated by rubbing home cage bedding on walls and floor of apparatus. Animals are tracked using IR sensors on the shuttle box. Data is collected and analyzed using the Graphic State, and Graphic State RT programs from Colbourn Instruments. Each experiment was performed over 4 consecutive days. Expressing $rd1$ mice are injected with $BGAG_{12,460}$ or $BGAG_{12,460}$ and $MAG0_{460}$ 24 h before entering the apparatus. Day 1–Animals are placed in dark shuttle cage (Colbourn Instruments, H10-11M-SC) isolated from noise, smell or peripheral light by the Habitest Isolation cubicle (Colbourn Instruments, H10-24). Animals are allowed to explore freely both sides of shuttle cage until they reach a stage in which they are exploring each side of the cage equally (15-45 min). Day 2–Animal is placed randomly on one side of the shuttle cage, and monitoring and illumination is begun after crossing the decision point. Upon entering adverse side of the cage, a 2 s, 0.7 mA shock was given after 5 s of entry and at 5 s intervals until the animal returns to the side of the cage associated with the non-aversive stimuli or after 60 s on this side and reported as a failed trial (resulting in a repeated run after an adequate resting period). All trials last 15 minutes from the time the animal first enters the aversive side. (Colbourn Instruments, H13-15-Precision shocker) Day 3–Repeat training as described in Day 2. Day 4–Light stimuli was reversed (to avoid a bias for location rather than pattern) and flooring of cage is replaced with plastic instead of shock grating. This ensures that the only related association with training day is the light stimuli. Animals are then assessed for length of time they spend on the side of the aversive stimuli vs. the safe side. Trial is 15 min in length from the time the animal enters the aversive compartment. Due to the sensitivity of $wt$ mice, the intensity of the LED arrays were reduced to 0.5 mW cm$^{-2}$. Visual discrimination optical angle calculations were performed using the parameters of the behavioral shuttle cage ($15.24 \times 17.76$ cm$^2$), the distance from decision point (divider) and the mounted LED panel (18.85 cm), and the parameters of the stimulus pattern (6–1 cm between parallel lines) using the optical (physical) angle equation. Conversion from visual angles to Snellen values was achieved using a conversion chart for human patients found in Holladay[59].

**Data availability**. The data that support the findings of this study are available from the corresponding author upon reasonable request.

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

## Acknowledgements

We thank Leah Byrne, Cameron Baker, Prashant Donthamsetti, Autoosa Salari and Richard Kramer for helpful discussion and Adam Hoagland, Zhu Fu, Elizabeth Carroll, Drew Friedmann, Aleksandra Polosukhina, and Ivan Tochitsky for technical assistance. The work was supported by the National Institutes of Health Nanomedicine Center for the Optical Control of Biological Function (2PN2EY018241) (E.Y.I., J.F. and D.T.) and instrumentation award (S10 RR028971) and the National Science Foundation Major Research Instrumentation Award (1041078) and EAGER award (IOS-1451027) (E.Y.I.). We are grateful to the Centre for Integrated Protein Science Munich (CIPSM) (J.B. and D.T.), the 'EPFL Fellows' fellowship programme co-funded by Marie Skłodowska-Curie, Horizon 2020 (665667) (J.B.) and an Advanced Grant from the European Research Council (268795) (D.T.). We kindly thank Prof. Dr. Kai Johnsson for providing BG- and BC-containing compounds.

## Author contributions

M.H.B., A.H., J.L., and B.M.G. designed and performed experiments and analyzed data, with input from J.B., J.F., D.T., and E.Y.I. Chemical synthesis was designed and performed by J.B. with input from D.T. Extended release with beta cyclodextrin developed by K.C. and R.H.K. Behavioral experiments were performed by A.H. Viral synthesis, cloning, analytic tools, and statistical consultation provided by M.V., C.S., Y.J.K., and K. A. The paper was written by M.H.B. and E.Y.I. with input from all of the authors.

## Additional information

**Competing interests:** E.Y.I. is a co-founder of Photoswitch Therapeutics, which is developing approaches to vision restoration that may include use of the system described here. The remaining authors declare no competing financial interests.

