## [Peer Review File · Nature Communications]

Reviewers' comments:

Reviewer #1 (Remarks to the Author):

In their study, Berry et al. deliver a modified GPCR (SNAP-mGluR2), which this group of investigators recently generated and characterized in HEK293 cells (Levitz et al, PNAS 2017), via adeno-associated viruses (AAVs) to retinal ganglion cells (RGCs) in mice with photoreceptor degeneration (rd1 mice). They "charge" receptors by conjugation to a photoswitch (BGAG; SNAG-mGluR2), and show that this endows RGCs with rapid OFF responses (i.e. increase in firing to light decrements) and rapid ON suppression (i.e. decrease in firing to light increments). Because of the relatively fast kinetics of SNAG-mGluR2s, RGC firing is able to follow flicker stimuli up to 8 Hz. The authors show that SNAG-mGluR2s charged in vivo by intraocular injection of BGAG restore photophobia to rd1 mice, that effects from a single BGAG-injection persist for approximately 2 weeks, and that this can be extended to approximately 4 weeks by injecting BGAG in cyclodextrin. Importantly, the authors show that SNAG-mGluR2s can restore pattern vision to rd1 mice, achieving higher acuity than LiGluR, a light-activated designer channel that endows RGCs with ON responses. Finally, the authors show that co-injection of AAVs encoding LiGluR and SNAP-mGluR2 enhances the diversity of RGC light responses (ON sustained, ON transient, ON-OFF, and OFF) and further improves acuity in rd1 mice.

Thus, this study characterizes a new tool for vision restoration with a number of exciting properties (OFF light responses, fast response kinetics, long-lasting photocharging). The authors convincingly demonstrate the ability of this tool to restore pattern discrimination to blind mice and show that combinatorial use with tools that restore ON responses further improves vision restoration. The experiments appear to be carefully conducted, are explained clearly in the text, and are illustrated well in the figures. However, in my opinion, additional experiments clarifying how SNAG-mGluR2 restores light responses to RGCs should be performed.

Specific comments

1) RGCs in mice with retinal degeneration (incl. rd1) are spontaneously hyperactive. It is possible that OFF responses generated by SNAG-mGluR2 depend on this increase in baseline firing, which is suppressed during the ON phase of the stimulus with a subsequent rebound at light OFF. The authors should test the effect of SNAG-mGluR2 in wild-type mice when transmission of photoreceptor signals to bipolar cells is blocked by L-APB and ACET.

2) It is unclear what conductances are activated by SNAG-mGluR2 in RGCs. This question could be addressed by patch clamp recordings from RGCs and/or pharmacology.

Reviewer #2 (Remarks to the Author):

This is a very well-conceived set of experiments in a manuscript that is a pleasure to read.

The authors present solid evidence for a photo-engineered G protein coupled receptor with and without a LiGluR that restores light evoked spiking activity to retinal ganglion cells in rd1 mice. The novelty of this approach is the new GPCR that is used that demonstrates superior performance in both sensitivity, timing and duration of effect.

The results are clearly presented with for the most part all controls in place. There are some minor issues with the statistics that are used. Finally, because it has been shown that some of these channels only function in "blind" retinas, it would be useful to know if these receptors function in the WT retina in which synaptic function has been eliminated pharmacologically.

Line 48 – of retina is redundant

Line 54 – this dual system improve close-line discrimination. What is the comparison?

Line 98 "that would turn it on in response to light" – phrase is odd and it is not clear what it means

Line 144 – 145 localized to the somata and dendrites of both ON and OFF RGCs. – There is no data that definitively shows dendritic label, what the authors show is label throughout the IPL.

Line 157 – Figure 1h would be even more convincing if raw data were shown for the condition where LY341495 were included.

Figure 1g – The solid bars obscure the response decrease. An outline of the bar would be more appropriate.

Figure 1h – how many cells in the number of retinas described? Why is there no statistical comparison? If one is made, the authors need to remember that here as in many of the figures, it is not appropriate to perform a parametric statistic (t-test, ANOVA) on data that are computed as a proportion/percentage, as these data are not normally distributed.

Line 166 – loses should be lost

Line 180 – 500x of what?

Figure 2f – please provide statistical comparison

Figure 2g, h – The most appropriate analysis here would be a fast fourier transform to examine the power at the frequency of the stimulus. Then a statistical comparison should be made.

In addition, the authors should keep in mind when they plot the inhibitory response that there is a floor effect when using spiking responses, there could be much more inhibition in terms of the cell's membrane potential, but spiking can only go to 0sp/sec.

Figure 3 b Stat should be a repeated measure ANOVA.

Figure 3e – f Stat should be a nonparametric ANOVA

Figure 4K - Stat should be a nonparametric ANOVA

Line 289 – shouldn't the index be a cross correlation index?

Figure 5i - - Stat should be a nonparametric ANOVA

Discussion – It would be useful if the authors discussed how they think that this mGluR2 receptor creates the change in the cell polarization. Could they speculate on what they think might be the downstream target of the receptor and what GPCR cascade is being used.

We thank the reviewers for their helpful comments. We have addressed all of the questions that were raised, as described below. To do so, we added new experiments, analysis and expanded our explanation and interpretation of the results (highlighted yellow in the text and supplement).

Reviewer #1

In their study, Berry et al. deliver a modified GPCR (SNAP-mGluR2), which this group of investigators recently generated and characterized in HEK293 cells (Levitz et al, PNAS 2017), via adeno-associated viruses (AAVs) to retinal ganglion cells (RGCs) in mice with photoreceptor degeneration (rd1 mice). They "charge" receptors by conjugation to a photoswitch (BGAG; SNAG-mGluR2), and show that this endows RGCs with rapid OFF responses (i.e. increase in firing to light decrements) and rapid ON suppression (i.e. decrease in firing to light increments). Because of the relatively fast kinetics of SNAG-mGluR2s, RGC firing is able to follow flicker stimuli up to 8 Hz. The authors show that SNAG-mGluR2s charged in vivo by intraocular injection of BGAG restore photophobia to rd1 mice, that effects from a single BGAG-injection persist for approximately 2 weeks, and that this can be extended to approximately 4 weeks by injecting BGAG in cyclodextrin. Importantly, the authors show that SNAG-mGluR2s can restore pattern vision to rd1 mice, achieving higher acuity than LiGluR, a light-activated designer channel that endows RGCs with ON responses. Finally, the authors show that co-injection of AAVs encoding LiGluR and SNAP-mGluR2 enhances the diversity of RGC light responses (ON sustained, ON transient, ON-OFF, and OFF) and further improves acuity in rd1 mice.

Thus, this study characterizes a new tool for vision restoration with a number of exciting properties (OFF light responses, fast response kinetics, long-lasting photocharging). The authors convincingly demonstrate the ability of this tool to restore pattern discrimination to blind mice and show that combinatorial use with tools that restore ON responses further improves vision restoration. The experiments appear to be carefully conducted, are explained clearly in the text, and are illustrated well in the figures. However, in my opinion, additional experiments clarifying how SNAG-mGluR2 restores light responses to RGCs should be performed.

We thank the reviewer for the positive overview and address the specific issues raised below.

Specific comments

1) RGCs in mice with retinal degeneration (incl. rd1) are spontaneously hyperactive. It is possible that OFF responses generated by SNAG-mGluR2 depend on this increase in baseline firing, which is suppressed during the ON phase of the stimulus with a subsequent rebound at light OFF. The authors should test the effect of SNAG-mGluR2 in wild-type mice when transmission of photoreceptor signals to bipolar cells is blocked by L-APB and ACET.

We have performed the suggested experiment. It appears that SNAG-mGluR2 light response has specificity for the degenerated retina.

SNAP-mGluR2 was virally delivered to RGCs of *wt* animals as we did in *rd1* animals. These retinas displayed a pan-retinal expression pattern that was restricted to the RGC layer (Supplementary Fig. 4a-c), as seen in the *rd1* retina (Fig. 1c,d). Retinas mounted on the MEA and conjugated to BGAG_{12,460} had low basal activity in the dark and normal photoreceptor mediated ON and OFF responses when flashed with dim white light that is bright enough to elicit a response from the *wt* untreated retina, but below the threshold of SNAG-mGluR2 (Fig. 1j,k & Supplementary Fig. 3j). Administration of 50 μ M of L-AP4 (agonist of mGluR6¹), and 1 μ M of ACET (blocker of kainate receptors²), which mediate synaptic transmission from photoreceptor cells to ON & OFF bipolar cells, abolished this photoreceptor-driven light response (Fig. 1j,k & Supplementary Fig. 3k). Strikingly, illumination at the an intensity (20 mW/cm²) and wavelength (445 nm) that produces a large SNAG-mGluR2 light response in the RGCs of the *rd1* retina elicited little to no light response in the synaptic transmission-blocked *wt* retina (Fig. 1j,k). When light intensity was increased by 2.5-fold a small light response was detected that was ~7x smaller than that induced in *rd1* animals expressing SNAG-mGluR2 and labeled with BGAG (Fig. 1k & Supplementary Fig. 3l). The mechanism for this selectivity in response may depend, as reviewer #1 and reviewer #2 suggested, on the increase in baseline firing seen in *rd1* animals^{3,4}. Additionally, the reported increase in Kv7 channel activity and up-regulation of HCN channels in the degenerating retina may also contribute to selective function in the disease model⁵⁻⁷. Regardless, these results suggests that the signaling of SNAG-mGluR2 occurs selectively (or far more robustly) in

the degenerating retina, and that, in early stage of the disease, when only some of the photoreceptors have been lost, this therapy may selectively animate RGCs whose photoreceptor cells have degenerated, with minimal effect on areas whose photoreceptor cells remain intact.

2) It is unclear what conductances are activated by SNAG-mGluR2 in RGCs. This question could be addressed by patch clamp recordings from RGCs and/or pharmacology.

We used pharmacology on the multi-electrode array in effort to identify the downstream target of mGluR2 in RGCs of the *rd1* retina. “As expected for the Gi-coupled mGluR2⁸, the light response was absent in retina injected with 150 uM pertussis toxin for 24 hrs (Fig. 1h & Supplementary Fig. 3c). We considered that the inhibitory response could be mediated by activation of a potassium channel. We first tested G protein-coupled inwardly-rectifying potassium (GIRK) channels, since these channels are expressed in RGCs⁹ and are activated by Gβγ as a result of activation of mGluR2¹⁰. However, neither 300 nM Tertiapin-Q nor 1mM barium, potent blockers of GIRK channels¹¹, altered the SNAG-mGluR2 light response in the *rd1* retina (Fig. 1i & Supplementary Fig. 3e-g). We next considered Kv7 channels, since one of these, Kv7.4, is also activated by Gβγ^{12,13} and is expressed in the retina^{14,15}. We found that 500 nM linopirdine, a potent and, at this low concentration, selective blocker of Kv7 channels¹⁶, blocked the SNAG-mGluR2 light response in the *rd1* retina (Fig. 1j & Supplementary Fig. 3h). This suggests that the SNAG-mGluR2 inhibitory light response is mediated by activation of the Kv7.4 channel. Lastly, we tested the idea that SNAG-mGluR2 OFF response could be due to hyperpolarization-activated cyclic nucleotide-gated HCN channels, since they are expressed in RGCs¹⁷ and have been established in the mechanism of hyperpolarization-induced rebound excitation within RGCs¹⁸. Indeed, we found that 50 μM ivabradine, a blocker of HCN channels¹⁹, eliminated the excitatory OFF response while maintaining the inhibitory component when the light was on (Fig. 1j & Supplementary Fig. 3i). Together, these results suggest that SNAG-mGluR2 activates Kv7.4 channels through Gi-linked Gβγ coupling to hyperpolarize RGCs and that deactivation of the channels when the light is turned off triggers rebound excitation due to activation of HCN channels.

Reviewer #2

*This is a very well-conceived set of experiments in a manuscript that is a pleasure to read. The authors present solid evidence for a photo-engineered G protein coupled receptor with and without a LiGluR that restores light evoked spiking activity to retinal ganglion cells in *rd1* mice. The novelty of this approach is the new GPCR that is used that demonstrates superior performance in both sensitivity, timing and duration of effect.*

We thank the reviewer for this positive assessment.

The results are clearly presented with for the most part all controls in place. There are some minor issues with the statistics that are used. Finally, because it has been shown that some of these channels only function in “blind” retinas, it would be useful to know if these receptors function in the WT retina in which synaptic function has been eliminated pharmacologically.

We have performed the suggested experiment and it appears that SNAG-mGluR2 OFF response provides some specificity for degenerative retina. Please see the response to specific comment #1 by Reviewer #1, who posed the same question.

Line 48 – of retina is redundant

Removed.

Line 54 – this dual system improve close-line discrimination. What is the comparison?

Changed to: This dual system improves close-line discrimination over either on its own.

Line 98 “that would turn it on in response to light” – phrase is odd and it is not clear what it means

Changed to: To obtain a native retinal GPCR that endows a rapid light response onto RGCs, we engineered metabotropic glutamate receptor 2 (mGluR2) to contain a nanoscopic chemical photoswitch that activates mGluR2 in response to light.

Line 144 – 145 localized to the somata and dendrites of both ON and OFF RGCs. – There is no data that definitively shows dendritic label, what the authors show is label throughout the IPL.

Changed to: and covering the area where the somata and dendrites of both ON- and OFF-RGCs are located.

Line 157 – Figure 1h would be even more convincing if raw data were shown for the condition where LY341495 were included.

We have followed the reviewer’s suggestion and added Supplementary Fig. 3a,b which displays a representative MEA recording from *rd1* mouse retina expressing SNAG-mGluR2 in RGCs before (a) and after (b) addition of 5 μ M LY341495 (a,b: n=71) We have also included traces of MEA recordings as a supplement to the additional experiments suggested by the reviewer (Supplementary Fig. 3).

Figure 1g – The solid bars obscure the response decrease. An outline of the bar would be more appropriate.

Changed to small bar above denoting duration of illumination.

Figure 1h – how many cells in the number of retinas described? Why is there no statistical comparison? If one is made, the authors need to remember that here as in many of the figures, it is not appropriate to perform a parametric statistic (t-test, ANOVA) on data that are computed as a proportion/percentage, as these data are not normally distributed.

Number of cells has been added to 1h. Additionally we have addressed the reviewer’s comments regarding statistical comparisons. All statistical comparisons have been recalculated using nonparametric tests (Supplementary Tables 1 and 4).

Figure 1h: To determine significance between drug treatments in cells before and after exposure, a Mann-Whitney U test was applied to the data.

	Mann-Whitney U test (two tailed)	
		Significance?
Rd1 SNAP-mGluR2		
BGAG (Fig. 1h)		
(-) BGAG inhibition vs (+) BGAG inhibition	p<0.001	Yes
(-) BGAG OFF response vs (+) BGAG OFF response	p<0.001	Yes
Rd1 SNAG-mGluR2		
LY34 (Fig. 1h)		
(-) drug inhibition vs (+) drug inhibition	p<0.001	Yes
(-) drug OFF response vs (+) drug OFF response	p<0.001	Yes
PTX (Fig. 1h)		
(-) drug inhibition vs (+) drug inhibition	p<0.001	Yes
(-) drug OFF response vs (+) drug OFF response	p<0.001	Yes

Line 166 – loses should be lost
Corrected

Line 180 – 500x of what?

Corrected “suggesting a therapeutic window of at least 500x (tolerated dose / dose required for maximal efficacy = 250 μ M / 0.5 μ M)”.

Figure 2f – please provide statistical comparison

The statistical comparison was between the light induced inhibition and OFF response of *rd1* SNAG-mGluR2 retina to that of *rd1* untreated retina.

Figure 2g, h – The most appropriate analysis here would be a fast fourier transform to examine the power at the frequency of the stimulus. Then a statistical comparison should be made.

In addition, the authors should keep in mind when they plot the inhibitory response that there is a floor effect when using spiking responses, there could be much more inhibition in terms of the cell's membrane potential, but spiking can only go to 0sp/sec.

While an FFT analysis, as for example performed recently to determine the temporal sensitivity of cone and rod mediated responses in *wt* mouse RGCs²⁰, would provide an interesting comparison between the natural photoreceptor-mediated responses and responses mediated by SNAG-mGluR2 expressed in the RGCs of *rd1* mice, it is beyond the scope of this study. Here we did not set out to determine peak temporal sensitivity, but instead only endeavored to estimate following frequency, and so our experiments with trains of light pulses simultaneously varied both light pulse frequency and duration.

We agree with the reviewer that there will be a floor effect when spike responses hit zero, while the cell membrane potential may be further hyper-polarized. Indeed, we see spike trains from individual units in several of the figures in which early quiescence during the start of the light pulse relaxes to a low rate of firing, as the average measured spiking rate hits a minimum early and relaxes up. This means that we underestimate the degree of inhibition to some degree under conditions where the inhibition is the strongest, distorting the relationship between pulse duration and inhibition. We now address this in Methods.

Figure 3 b Stat should be a repeated measure ANOVA.

We have followed the reviewer's suggestion and performed a repeated measure ANOVA. Statistical significance calculations for slow release BGAG delivery for behavioral light avoidance (Fig. 3b) was analyzed by repeated-measures ANOVA (rANOVA) using "time point" as the within-subject variable and "group" (control untreated, treated) as the between-subject variable. Within-subject effects were analyzed by one-way ANOVA using "time point" as the independent variable. Where sphericity was violated, as assessed by Mauchly's test of sphericity, the Greenhouse–Geisser correction was applied (Supplementary Table 2).

repeated measure ANOVA			
(days 7-42)	sphericity assumed	Greenhouse-Geisser correction	
Rd1 SNAP-mGluR2 with cyclodextrin + BGAG	p = 0.418	p = 0.401	Statistically insignificant
Rd1 untreated control	p = 0.622	p = 0.510	Statistically insignificant
Between subject one-way ANOVA			
Rd1 SNAP-mGluR2 with cyclodextrin + BGAG vs Rd1 untreated control		p < 0.005 for days 7-42	Statistically significant

So the statistical difference was not significant within subjects across multiple days but was statistically significant between the two groups. This suggests that treated mice show an avoidance advantage over *rd1* control animals and display no change in performance over 42 days. This significance was lost at day 48 where treated animals performed no differently from control *rd1* animals.

Figure 3e – f Stat should be a nonparametric ANOVA

Corrected. See Supplementary Fig. 5b,d-g

The behavioral performance of all 156 mice that participated in learned avoidance behavior (Fig. 3e-h & Fig. 5i) was reassessed as proportion of success on the behavioral task; an approach commonly employed in nonparametric calculations of clinical trials or behavioral experiments where achieving a normal distribution is unrealistic. For learned dark avoidance behavior Fig. 3e and the learned pattern discrimination behaviors (Fig. 3f-h & 5i), a standard deviation was computed. A success was defined as greater than the sum of the control group average and one S.D, and a failure was any value that did not achieve this criteria. Success ratios were then calculated for each condition

(Supplementary Fig. 5b,d-g). To determine significance in differences between conditions a pairwise contingency table was then constructed, and a Two-Sided Pearson's Chi-Square Test was initially conducted (Supplementary Table 3). To correct for conditions with a small n, a One-Sided Fisher's Exact Test was also conducted (Supplementary Table 3). In addition to these tests, we request that the behavioral performance analysis on Fig. 3e-h using parametric (ANOVA/t-test) and corrections (Bonferroni) also remain in order to maintain consistency with the behavioral measures of other vision restoration papers.

	Pearson's Chi-Square Test (2 sided)	Fisher's Exact Test (1 sided)	Significance?
Dark averse (Fig. S5d & Fig. 3e)			
Rd1 SNAG-mGluR2 vs rd1 control	0.008	0.003	Yes
Perp. vs. parallel bars (Fig. S5e & Fig. 3f)			
Rd1 SNAG-mGluR2 vs rd1 LiGluR	0.814	0.65	No
Rd1 SNAG-mGluR2 vs rd1 untreated control	0.002	0.004	Yes
Rd1 SNAG-mGluR2 vs wt c57	0.352	0.51	No
Rd1 LiGluR vs rd1 untreated control	0.023	0.045	Yes
Rd1 LiGluR vs wt c57	0.292	0.5	No
Rd1 untreated control vs wt c57	0.003	0.008	Yes
Line differentiation (Fig. S5f Fig. 3g)			
Rd1 SNAG-mGluR2 vs rd1 LiGluR	0.348	0.371	No
Rd1 SNAG-mGluR2 vs rd1 untreated control	<0.001	0.001	Yes
Rd1 SNAG-mGluR2 vs wt c57	0.551	0.489	No
Rd1 LiGluR vs rd1 untreated control	<0.001	<0.001	Yes
Rd1 LiGluR vs wt c57	0.133	0.202	No
Rd1 untreated control vs wt c57	0.007	0.017	Yes
Close line differentiation (Fig. S5g,h Fig. 3h & 5i)			
Rd1 SNAG-mGluR2 vs rd1 untreated control	0.025	0.03	Yes
Rd1 SNAG-mGluR2 vs wt c57	0.732	0.534	No
Rd1 LiGluR vs rd1 untreated control	0.411	0.343	No
Rd1 LiGluR vs wt c57	0.098	0.104	No
Rd1 untreated control vs wt c57	0.022	0.03	Yes
Rd1 SNAG-mGluR2 vs rd1 combo.	0.007	0.01	Yes
Rd1 LiGluR vs rd1 combo.	<0.001	<0.001	Yes
Rd1 untreated control vs rd1 combo.	<0.001	<0.001	Yes
wt c57 vs rd1 combo.	0.019	0.045	Yes

Figure 5i - - Stat should be a nonparametric ANOVA

Corrected: Please see response, table above & Supplementary Fig. 5g

Figure 4K - Stat should be a nonparametric ANOVA

Corrected - To determine significance between cross correlation values, a Mann-Whitney U test was applied to the data.

Mann-Whitney U test (two tailed)

		Significance?
cross correlation (Fig. 4k)		
Rd1 SNAG-mGluR2 vs wt c57	p<0.001	Yes
Rd1 SNAG-mGluR2 vs rd1 combo.	p<0.001	Yes
Rd1 LiGluR vs wt c57	p<0.001	Yes
Rd1 LiGluR vs rd1 combo.	p<0.001	Yes

Line 289 – shouldn't the index be a cross correlation index?

Correct, changed to cross-correlation index

Discussion – It would be useful if the authors discussed how they think that this mGluR2 receptor creates the change in the cell polarization. Could they speculate on what they think might be the downstream target of the receptor and what GPCR cascade is being used.

Our results suggest that SNAG-mGluR2 activates Kv7.4 channels through Gi-linked G $\beta\gamma$ coupling to hyperpolarize RGCs and that deactivation of the channels when the light is turned off triggers rebound excitation due to activation of HCN channels. Please see the response to specific comment #2 by Reviewer #1, who posed the same question.

References for Reviewer Comments

1. Selvam, C., Goudet, C., Oueslati, N., Pin, J.P. & Acher, F.C. L-(+)-2-Amino-4-thiophosphonobutyric acid (L-thioAP4), a new potent agonist of group III metabotropic glutamate receptors: increased distal acidity affords enhanced potency. *Journal of medicinal chemistry* **50**, 4656-4664 (2007).
2. Dargan, S.L., *et al.* ACET is a highly potent and specific kainate receptor antagonist: characterisation and effects on hippocampal mossy fibre function. *Neuropharmacology* **56**, 121-130 (2009).
3. Goo, Y.S., *et al.* Spontaneous Oscillatory Rhythm in Retinal Activities of Two Retinal Degeneration (rd1 and rd10) Mice. *The Korean journal of physiology & pharmacology : official journal of the Korean Physiological Society and the Korean Society of Pharmacology* **15**, 415-422 (2011).
4. Trenholm, S. & Awatramani, G.B. Origins of spontaneous activity in the degenerating retina. *Frontiers in Cellular Neuroscience* **9**, 277 (2015).
5. Caminos, E., Vaquero, C.F. & Martinez-Galan, J.R. Relationship between rat retinal degeneration and potassium channel KCNQ5 expression. *Experimental Eye Research* **131**, 1-11 (2015).
6. Tochitsky, I., *et al.* How Azobenzene Photoswitches Restore Visual Responses to the Blind Retina. *Neuron* **92**, 100-113.
7. Stradleigh, T.W., *et al.* Colocalization of hyperpolarization-activated, cyclic nucleotide-gated channel subunits in rat retinal ganglion cells. *The Journal of comparative neurology* **519**, 2546-2573 (2011).
8. Kammermeier, P.J., Davis, M.I. & Ikeda, S.R. Specificity of metabotropic glutamate receptor 2 coupling to G proteins. *Molecular pharmacology* **63**, 183-191 (2003).
9. Clark, B.D., Kurth-Nelson, Z.L. & Newman, E.A. Adenosine-evoked hyperpolarization of retinal ganglion cells is mediated by G-protein-coupled inwardly rectifying K⁺ and small conductance Ca²⁺-activated K⁺ channel activation. *J Neurosci* **29**, 11237-11245 (2009).
10. Raveh, A., Cooper, A., Guy-David, L. & Reuveny, E. Nonenzymatic Rapid Control of GIRK Channel Function by a G Protein-Coupled Receptor Kinase. *Cell* **143**, 750-760 (2010).
11. Dascal, N., *et al.* Expression of an atrial G-protein-activated potassium channel in *Xenopus* oocytes. *Proc Natl Acad Sci U S A* **90**, 6596-6600 (1993).
12. Povstyan, O.V., Barrese, V., Stott, J.B. & Greenwood, I.A. Synergistic interplay of G $\beta\gamma$ and phosphatidylinositol 4,5-bisphosphate dictates Kv7.4 channel activity. *Pflügers Archiv - European Journal of Physiology* **469**, 213-223 (2017).
13. Stott, J.B., Povstyan, O.V., Carr, G., Barrese, V. & Greenwood, I.A. G-protein betagamma subunits are positive regulators of Kv7.4 and native vascular Kv7 channel activity. *Proc Natl Acad Sci U S A* **112**, 6497-6502 (2015).
14. Beisel, K.W., *et al.* Differential expression of KCNQ4 in inner hair cells and sensory neurons is the basis of progressive high-frequency hearing loss. *J Neurosci* **25**, 9285-9293 (2005).
15. Zhang, X., Yang, D. & Hughes, B.A. KCNQ5/K(v)7.5 potassium channel expression and subcellular localization in primate retinal pigment epithelium and neural retina. *American Journal of Physiology - Cell Physiology* **301**, C1017-C1026 (2011).
16. Pattnaik, B.R. & Hughes, B.A. Effects of KCNQ channel modulators on the M-type potassium current in primate retinal pigment epithelium. *American Journal of Physiology - Cell Physiology* **302**, C821-C833 (2012).
17. Stradleigh, T.W., *et al.* Colocalization of HCN Channel Subunits in Rat Retinal Ganglion Cells. *The Journal of comparative neurology* **519**, 2546-2573 (2011).
18. Mitra, P. & Miller, R.F. Mechanism underlying rebound excitation in retinal ganglion cells. *Visual neuroscience* **24**, 709-731 (2007).

19. Bucchi, A., Tognati, A., Milanesi, R., Baruscotti, M. & DiFrancesco, D. Properties of ivabradine-induced block of HCN1 and HCN4 pacemaker channels. *The Journal of physiology* **572**, 335-346 (2006).
20. Wang, Y.V., Weick, M. & Demb, J.B. Spectral and temporal sensitivity of cone-mediated responses in mouse retinal ganglion cells. *The Journal of neuroscience : the official journal of the Society for Neuroscience* **31**, 7670-7681 (2011).

REVIEWERS' COMMENTS:

Reviewer #1 (Remarks to the Author):

The authors have satisfactorily addressed my previous concerns. Congratulations on a very nice study.

Reviewer #2 (Remarks to the Author):

In this resubmitted manuscript, the authors have responded to all of the critiques of the original submission.

The authors report on a very exciting finding that may have far reaching relevance, not only in the field of therapeutics but also in basic research.

It also was a pleasure to read.

I have only a few minor comments or found errors that I want to point out for the authors.

Page 6 the authors should be consistent about their reference to mGluR2 or mGLuR2.
Line 155 – I do not think that the sentence refers to Fig 1e.

Page 7 - The abbreviation that the authors use for Pertussis toxin, PTX also is a common abbreviation for picrotoxin, a more frequently use drug in retinal circuit research. I suggest that the consider writing out pertussis toxin or at least write it out on first use in the figures.

Lines 168 – 187. References to Figure 1i and j seem to be mixed up a number of times.
Line 184, I do not see what the authors are alluding to in the text in Figure 3i.

Page 8 – line 205 I think that the authors are describing Figure2j and not 2i as indicated.

Page 9 – computation for the time to peak suppression of the ON response is provided by not or the OFF response. It is interesting to know what the timing of the response onset is and a comparison to WT OFF responses would improve this portion of the manuscript.

When I look at the response peaks as a function of intensity, I do not see a response that is 20% of the maximum in the figure provided. It looks closer to 12%. Is this the selection of the response that is not emblematic of the mean or is do the authors include the inhibition in the computation. I would think that it would be more logical to compute from baseline for each of the responses. The also is the possibility that I did not read the methods carefully enough.

Page 10 and page 14. The authors use the terms incipient and excipient in the same way. Probably only one is correct.

Page 10 line 255. The term installed is odd in this context.

Page 13 – the authors use the term LRI, I believe for the first time. While defined in the methods, one does not get to the methods until after reading the results, suggest defining the term (not describing the methods).

Page 14 – line 395. Intestines should be intensities. (LOL).

Page 15 – when the two vectors were used in combination, did the spontaneous activity change in the rd1 RGCs?

We thank the reviewers and the editor for their supportive comments. We have addressed all of the questions and fixed corrections that were raised, as described below and made the necessary changes noted by the editor in the supplementary text. Changes have been tracked and highlighted (yellow) in the main text.

Reviewer #1 (Remarks to the Author):

The authors have satisfactorily addressed my previous concerns. Congratulations on a very nice study.

Reviewer #2 (Remarks to the Author):

In this resubmitted manuscript, the authors have responded to all of the critiques of the original submission.

The authors report on a very exciting finding that may have far reaching relevance, not only in the field of therapeutics but also in basic research.

It also was a pleasure to read.

I have only a few minor comments or found errors that I want to point out for the authors.

Page 6 the authors should be consistent about their reference to mGluR2 or mGLuR2. Line 155 – I do not think that the sentence refers to Fig 1e.

Corrected to mGluR2 throughout

Page 7 - The abbreviation that the authors use for Pertussis toxin, PTX also is a common abbreviation for picrotoxin, a more frequently used drug in retinal circuit research. I suggest that they consider writing out pertussis toxin or at least write it out on first use in the figures.

Changed to Pertussis toxin in main figure, legends, and text

Lines 168 – 187. References to Figure 1i and j seem to be mixed up a number of times.

Corrected

Line 184, I do not see what the authors are alluding to in the text in Figure 3i.

Inhibition present in raster. Average of 6 flashes shown here for the reviewer:

Page 8 – line 205 I think that the authors are describing Figure2j and not 2i as indicated.

Corrected

Page 9 – computation for the time to peak suppression of the ON response is provided by not or the OFF response. It is interesting to know what the timing of the response onset is and a comparison to WT OFF responses would improve this portion of the manuscript.

Time from light termination to peak OFF response was measured in individual cells across both *wt* retina and *rd1* retina expressing SNAG-mGluR2. *rd1* treated retina display a short delay in OFF response that is only slightly slower than *wt* retina with intact photoreceptors. The time to peak mean \pm SEM has been added to the text and figure legends.

When I look at the response peaks as a function of intensity, I do not see a response that

is 20% of the maximum in the figure provided. It looks closer to 12%. Is this the selection of the response that is not emblematic of the mean or is do the authors include the inhibition in the computation. I would think that it would be more logical to compute from baseline for each of the responses. The also is the possibility that I did not read the methods carefully enough.

Mean response (Fig. 2h) was 17% which we said was ~20% in the text. Corrected in the text as a range (10-20%) to account for variability in the response at 25ms. The peak inhibition and peak OFF response were both computed from baseline, this is now clarified in Methods.

Page 10 and page 14. The authors use the terms incipient and excipient in the same way. Probably only one is correct.

Corrected to excipient

Page 10 line 255. The term installed is odd in this context.

Changed to “sought to determine if blind mice restored with OFF light responses could perform visual tasks”

Page 13 – the authors use the term LRI, I believe for the first time. While defined in the methods, one does not get to the methods until after reading the results, suggest defining the term (not describing the methods).

Light response index (LRI) and photoswitching index (PSI) are the same thing and were used interchangeably before. They have all been changed to LRI and defined on first use.

Page 14 – line 395. Intestines should be intensities. (LOL).

Corrected.

Page 15 – when the two vectors were used in combination, did the spontaneous activity change in the rd1 RGCs?

Spontaneous activity is slightly decreased in some of the cases when the 2 vectors are co-expressed, however this effect is variable, so we are unable to make a conclusion.